# Eco-evolutionary dynamics of multigames with mutations

**Sourav Roy[1], Sayantan Nag Chowdhury[2], Prakash Chandra Mali[1], Matjaž Perc[3,4,5,6], Dibakar Ghosh[2]***

**1** Department of Mathematics, Jadavpur University, Kolkata, West Bengal, India, **2** Physics and Applied Mathematics Unit, Indian Statistical Institute, Kolkata, West Bengal, India, **3** Faculty of Natural Sciences and Mathematics, University of Maribor, Maribor, Slovenia, **4** Department of Medical Research, China Medical University Hospital, China Medical University, Taichung, Taiwan, **5** Alma Mater Europaea, Maribor, Slovenia, **6** Complexity Science Hub Vienna, Vienna, Austria

* dibakar@isical.ac.in

**Data Availability Statement:** All relevant data are within the paper.

**Funding:** M.P. was supported by the Javna Agencija za Raziskovalno Dejavnost RS (Grant Nos. P1-0403 and J1-2457).

## Abstract

Most environments favor defection over cooperation due to natural selection. Nonetheless, the emergence of cooperation is omnipresent in many biological, social, and economic systems, quite contrary to the well-celebrated Darwinian theory of evolution. Much research has been devoted to better understanding how and why cooperation persists among self-interested individuals despite their competition for limited resources. Here we go beyond a single social dilemma since individuals usually encounter various social challenges. In particular, we propose and study a mathematical model incorporating both the prisoner's dilemma and the snowdrift game. We further extend this model by considering ecological signatures like mutation and selfless one-sided contribution of altruist free space. The nonlinear evolutionary dynamics that results from these upgrades offer a broader range of equilibrium outcomes, and it also often favors cooperation over defection. With the help of analytical and numerical calculations, our theoretical model sheds light on the mechanisms that maintain biodiversity, and it helps to explain the evolution of social order in human societies.

## 1 Introduction

Numerous interdisciplinary researchers have long sought a way to understand how the tremendous biodiversity among species persists in nature despite the significant differences between them in terms of competitive capability. Darwinian evolution [1] always challenges the emergence of spontaneous social cooperation, as cooperators act entail an inherent cost for displaying altruism [2]. Self-interested defectors generally exploit these cooperators and hinder the maintenance of cooperation. This trivially raises the question of why cooperators act selflessly if only the fittest succeeds and how numerous forms of cooperative behavior sustain in nature ranging from a low-level microbial biological system to a high-level complex social system. We resort to the evolutionary game dynamical interaction given by a 2 × 2 pay-off matrix to solve this riddle. These classical games are capable of analyzing the economic and strategic

**Competing interests:** We have declared that no competing interests exist.

decisions of rational individuals [3]. An unprecedented spectrum of researchers have already focused overwhelmingly on the evolutionary game theory [4–6] to seek out the mechanisms that sustain and promote cooperation [7–14].

We construct a mathematical model on evolutionary multigames [15–27] by adopting two simple two-player games, viz. the prisoner's dilemma [28] (PD) and snowdrift game [29, 30] (SD), as metaphors for cooperation between unrelated rational individuals. Our choice of combining two distinct games is motivated by the fact that people may face different social complexities rather than a single one. People take care of problems based on their own perceptions. This real-life behavioral heterogeneity among individuals inspires us to examine the effect of multigames for understanding social diversity. Our theoretical approach is another effective way to explore the impact of the interaction between different cultural backgrounds on large-scale social ordering [31].

While defection is the ultimate profitable strategy in the classical PD game, irrespective of the co-player's decision, a player's best strategy in the SD game depends on the co-player's decision. Thus, we select two different games with two contrasting outcomes. The stable coexistence of both cooperators and defectors is the expected consequence in the SD game, resulting in the persistence of cooperative behavior. In comparison, cheaters are always encouraged to exploit the cooperative individuals in the set-up of the PD game. In our constructed model, unrelated individuals have their probabilistic selection to choose which game they want to play. We also consider the mutation [32–42] as an evolutionary mechanism that enables people to switch their respective strategies. We assume that it is hard to behave rationally under every circumstance. Hence, in our mathematical model, we incorporate a simplistic assumption that cooperators' and defectors' subpopulations can interchange their strategies with a fixed bidirectional mutation rate. To the best of our knowledge, the majority of the studies on multigames have been examined in the absence of mutation.

Apart from that, to reveal the interplay between ecological and evolutionary dynamics, we inspect the influence of free space in our mathematical model. In the context of complex systems, free space [43–49] proves to be a promising factor for offering significant consequences on diverse collective dynamics. We assume that each individual's birth rate depends not only on the respective average payoffs but are also proportional to the available free space. This free space will give them reproductive opportunities. The population of unrelated players gains a payoff due to the interaction of the multigames. We treat these payoffs as reproductive successes. Individuals with a higher payoff leave more offspring if sufficient empty spaces are available and are able to outcompete less successful ones. Free space will also contribute to the individual's average payoffs. We consider free space as an ecological variable donating selflessly to all players' fitness. However, free space never anticipates any benefit from others. This generous nature is surprising in a society of self-interested individuals. Still, free space provides reproductive opportunities to each player and loses their own identity to improve the fitness of other individuals. We consider such a selfless act of free space because helping others is a common practice among human beings and other animals. This tendency of unselfish concern for other people is a unique recipe for promoting cooperation and favors the survival of various species. The inclusion of such ecological dynamics into evolutionary multigames gives rise to a fascinating way to reveal the influence of eco-evolutionary dynamics [50–60] on decision-making and the evolution of cooperation.

The concurrence of ecological changes and rational players' evolution, depicted in the present article, prevails through the same time scale, rather than the typical assumption that ecological processes are much faster than evolutionary processes [61]. Our theoretical model uncovers how the combination of ecological dynamics and game dynamics is beneficial for maintaining cooperative behaviors under the influence of mutation and leads to the stable

coexistence of interacting competitors. Earlier, Nag Chowdhury et al. [8, 16] examined the influence of eco-evolutionary dynamics on cooperative behavior's emergence and evolution under a cooperation-supporting mechanism, viz. punishment. Nevertheless, our model exhibits the coexistence of cooperators and defectors in the absence of any such supporting mechanism. Moreover, individuals' birth rates in those Refs. [8, 16] depend only on their respective average payoffs. Apart from the addition of free space's altruistic behavior in the payoff matrix, we furthermore include the importance of available free space on the birth rates by assuming their birth rates to be proportional to the available space. The rest of the article is structured as follows: In Sec. (2), we discuss how we construct our mathematical model in detail. The following section (3) deals with the existence, uniqueness, and boundedness of the model, along with extensive numerical simulations and discussions. The section (4) presents the summary of our findings. Following that, we provide a brief discussion in the last section (5) on the challenges in this work's context that need to be addressed and are worth studying in the future.

## 2 The model

To start with, we consider a simplistic assumption that each individual has two distinct choices, viz. (i) cooperation (**C**) and (ii) defection (**D**). Even they can play any of the two possible games (a) PD game and (b) SD game. They can adopt the PD game with probability $p$, and alternatively, they interact with other individuals by playing the SD game with the complimentary probability $(1 - p)$. Both of these two-person games can be given by the following two payoff matrices $A$ and $B$ respectively, where

$$A = \begin{array}{c} \\ \mathbf{C} \\ \mathbf{D} \end{array}\begin{pmatrix} \mathbf{C} & \mathbf{D} \\ R_{PD} & S_{PD} \\ T_{PD} & P_{PD} \end{pmatrix} \text{ and } B = \begin{array}{c} \\ \mathbf{C} \\ \mathbf{D} \end{array}\begin{pmatrix} \mathbf{C} & \mathbf{D} \\ R_{SD} & S_{SD} \\ T_{SD} & P_{SD} \end{pmatrix}.$$

in which the entries portray the payoff accumulated by the players in the left.

Here, $R_{PD}$ and $R_{SD}$ contemplate the reward for mutual cooperation among two players in the respective PD and SD games. Similarly, both unrelated players receive the punishment $P_{PD}$ and $P_{SD}$ for mutual defection in the games PD and SD, respectively. An exploited cooperator gains the sucker's payoff $S_{PD}$ and $S_{SD}$, respectively in the PD and SD games when confronted by a defector. The mixed choice yields the defector temptations $T_{PD}$ and $T_{SD}$ to exploit a cooperator in the PD and SD games, respectively. The payoff ranking of these four-game parameters determines the two-person games. The conventional relative ordering for the PD game is $T_{PD} > R_{PD} > P_{PD} > S_{PD}$ [8, 16, 31] and $2R_{PD} > S_{PD} + T_{PD}$ [62]. Without loss of any generality, we choose $T_{PD} = \beta > 1$, $R_{PD} = 1$, $P_{PD} = \eta \in (0, 1)$, and $S_{PD} = 0$. Similarly, we choose $T_{SD} = \beta > 1$, $R_{SD} = 1$, $S_{SD} = 0$, and $P_{SD} = -\eta \in (-1, 0)$, maintaining the relative ordering $T_{SD} > R_{SD} > S_{SD} > P_{SD}$ for the standard SD game [8, 16, 31]. Thus, the payoff matrices $A$ and $B$ become

$$A = \begin{array}{c} \\ \mathbf{C} \\ \mathbf{D} \end{array}\begin{pmatrix} \mathbf{C} & \mathbf{D} \\ 1 & 0 \\ \beta & \eta \end{pmatrix} \text{ and } B = \begin{array}{c} \\ \mathbf{C} \\ \mathbf{D} \end{array}\begin{pmatrix} \mathbf{C} & \mathbf{D} \\ 1 & 0 \\ \beta & -\eta \end{pmatrix}.$$

Note that, in both of these games, mutual cooperation leads to the payoff $R_{PD}$ and $R_{SD}$, which is relatively higher than $P_{PD}$ and $P_{SD}$, which one defector receives when playing with a defector. Thus, cooperation always promises higher income than defection if both the rational players choose the same strategy. The difference between these two games' relative ordering leads to a contrasting scenario. In the SD game, the interaction between the cooperator and defector always promises a better income in terms of payoff than the interaction between two

defectors. A reverse reflection is observed in the case of the PD game thanks to the choice of such relative ranking of game parameters in the PD game. The interaction between two defectors in the PD game allows them to earn more than a cooperator encountering a defector. The switching between $P$ and $S$ in the relative ordering of both games thus produces noticeable unexpected consequences on the evolution of cooperation.

Since a player can decide which game they want to play, thus the final payoff matrix for the multigames looks like

$$E = pA + (1 - p)B = \begin{array}{c} \\ \mathbf{C} \\ \mathbf{D} \end{array} \begin{array}{c} \mathbf{C} \quad\quad \mathbf{D} \\ \begin{pmatrix} 1 & 0 \\ \beta & (2p - 1)\eta \end{pmatrix} \end{array}.$$

Here, $p \in [0, 1]$ is the probability of playing the PD game. Moreover, we consider free space **F** as an ecological variable, which contributes altruistically by helping others. Nevertheless, free space does not get any benefits by giving them reproductive opportunities. We incorporate this charitable role of free space by extending the $2 \times 2$ payoff matrix $E$ to the $3 \times 3$ payoff matrix $G$ as follows

$$G = \begin{array}{c} \\ \mathbf{C} \\ \mathbf{D} \\ \mathbf{F} \end{array} \begin{array}{c} \mathbf{C} \quad\quad \mathbf{D} \quad\quad \mathbf{F} \\ \begin{pmatrix} 1 & 0 & \sigma_1 \\ \beta & (2p - 1)\eta & \sigma_2 \\ 0 & 0 & 0 \end{pmatrix} \end{array}.$$

The matrix $G$ clearly reveals that the free space never earns any payoff for their selfless charitable act; however, it contributes a positive payoff $\sigma_1$ and $\sigma_2$ to the cooperators and the defectors, respectively. Since most of the game parameters (not all) lies within the closed interval $[0, 1]$, we assume, for the sake of feasible comparison, $\sigma_1$ and $\sigma_2$ both lies within the interval $[0, 1]$. When $\sigma_1$ and $\sigma_2$ are equal to zero, free space will not contribute anything to anyone. However, whenever $\sigma_1$ and $\sigma_2$ attain positive values, individuals gain an additional payoff from free space.

Inspired by the Malthusianism, we consider the following set of differential equations governing the changes in frequencies of cooperators and defectors as a function of time $t$

$$\begin{aligned} \dot{x} &= x[b_{\mathbf{C}} - d_{\mathbf{C}}], \\ \dot{y} &= y[b_{\mathbf{D}} - d_{\mathbf{D}}], \end{aligned} \tag{1}$$

where

$$\begin{cases} b_{\mathbf{C}} = \text{birth rate of cooperators}, \\ b_{\mathbf{D}} = \text{birth rate of defectors}, \\ d_{\mathbf{C}} = \text{death rate of cooperators}, \\ d_{\mathbf{D}} = \text{death rate of defectors}. \end{cases}$$

Here, $x$ and $y$ are the normalized densities of cooperators and defectors, respectively. Let $z$ be the available free space. Thus, we have

$$x + y + z = 1. \tag{2}$$

Relation (2) assures that by studying $x$ and $y$ alone, one can easily capture the dynamics of the two strategies. We assume the birth rates of each individual depends crucially on the

available free space as well as on their respective average fitness. Thus, we consider

$$b_{\mathbf{C}} = zf_{\mathbf{C}} = (1 - x - y)f_{\mathbf{C}},$$
$$b_{\mathbf{D}} = zf_{\mathbf{D}} = (1 - x - y)f_{\mathbf{D}}, \tag{3}$$

where $f_{\mathbf{C}}$ and $f_{\mathbf{D}}$ are average fitness of the cooperators and the defectors, respectively. The average payoff of cooperators and defectors can be determined using the payoff matrix G of the multigames, and the relation (2) as follows,

$$f_{\mathbf{C}} = x.1 + y.0 + z.\sigma_1 = (1 - \sigma_1)x - \sigma_1 y + \sigma_1,$$
$$f_{\mathbf{D}} = x.\beta + y.(2p - 1)\eta + z.\sigma_2 \tag{4}$$
$$= (\beta - \sigma_2)x + [(2p - 1)\eta - \sigma_2]y + \sigma_2.$$

Note that the average fitness of free space is

$$f_{\mathbf{F}} = x.0 + y.0 + z.0 = 0. \tag{5}$$

This is expected as free space does not gain anything for its benevolent nature. For simplicity, we further assume that all individuals die at a uniform and constant mortality rate $\xi \in (0, 1]$. Hence using the relations (3) and (4), our constructed model (1) transforms into

$$\dot{x} = x[(1 - x - y)\{(1 - \sigma_1)x - \sigma_1 y + \sigma_1\} - \xi],$$
$$\dot{y} = y[(1 - x - y)\{(\beta - \sigma_2)x + ((2p - 1)\eta - \sigma_2)y + \sigma_2\} - \xi]. \tag{6}$$

Now, we introduce a constant probability $\mu$ as a rate with which each individual mutates from one strategy to the others, in a continuous manner,

$$x \underset{\mu}{\overset{\mu}{\rightleftarrows}} y. \tag{7}$$

Relation (7) reflects that the mutation probability $\mu$ from the cooperators to the defectors is identical to the mutation rate from the defectors to the cooperators. So using the system (6), the well-mixed population under the influence of bidirectional mutation gives rise to the differential equations

$$\dot{x} = x[(1 - x - y)\{(1 - \sigma_1)x - \sigma_1 y + \sigma_1\} - \xi] + \mu(y - x),$$
$$\dot{y} = y[(1 - x - y)\{(\beta - \sigma_2)x + ((2p - 1)\eta - \sigma_2)y + \sigma_2\} - \xi] + \mu(x - y). \tag{8}$$

The system (8) contains seven different parameters. We summarize the necessary information about these parameters in (Table 1).

## 3 Results and discussions

### 3.1 Existence, uniqueness and positive invariance

Before investigating the model (8) using numerical simulations, we first prove the positive invariance of the proposed system (8). Clearly, the functions on the right-hand side of the differential Eq (8) are continuously differentiable and at the same time, locally Lipschitz in the first quadrant of $\mathbb{R} \times \mathbb{R}$. This ensures the existence and uniqueness of solutions for the model (8) with suitable initial values. Note that the initial conditions $(x_0, y_0)$ must lie within the domain $[0, 1] \times [0, 1]$ maintaining the inequality $0 \leq x_0 + y_0 \leq 1$, as they represent the

**Table 1. Parameters with their physical significance and domains: The first column represents the symbols used in the present manuscript to describe several parameters.** At the same time, the second column depicts what these parameters exactly mean. The set of possible input values from which the parameters can assume their values is given in the third column of the table. For further details, please see the main text.

| Parameters | Physical Significance | Domain |
|---|---|---|
| $\xi$ | Death Rate | (0, 1] |
| $\beta$ | The gain of a defector while interacting with a cooperator | (1, 2) |
| $\eta$ | The payoff for mutual defection | (0, 1) |
| $p$ | Probability of playing the PD game | [0, 1] |
| $\mu$ | Mutation probability | [0, 1] |
| $\sigma_1$ | The altruistic incentive of free space towards the cooperators | [0, 1] |
| $\sigma_2$ | The altruistic incentive of free space towards the defectors | [0, 1] |

frequencies of the two strategies. To determine the positivity of the proposed model (8), we write the system as follows,

$$
\dot{x} = x\psi_1,
$$
$$
\dot{y} = y\psi_2,
$$

$$(9)$$

where $\psi_1 = b_C - d_C - \mu + \frac{\mu y}{x}$ and $\psi_2 = b_D - d_D - \mu + \frac{\mu x}{y}$ are two integrable functions in the Riemannian sense. Solving Eq (9), we get

$$
x = c_1 \exp(\int \psi_1 dt),
$$
$$
y = c_2 \exp(\int \psi_2 dt),
$$

$$(10)$$

where $c_1$ and $c_2$ are the integrating constants depending on the initial densities $x_0$ and $y_0$. This proves that both $x$ and $y$ are non-negative. Now, to calculate the upper bound of $x + y$, we proceed as follows. The dynamical Eq (8) yield

$$
\frac{d}{dt}(x + y) = (1 - x - y)[x f_C + y f_D] - \xi(x + y)
$$

$$(11)$$

Since as per our previous analysis, $x + y \geq 0$ and the mortality rate $\xi$ is always positive, we have $\xi(x + y) \geq 0$. Thus, (11) reduces to

$$
\frac{d}{dt}(x + y) \leq (1 - x - y)[x f_C + y f_D].
$$

$$(12)$$

Integrating both sides, we get

$$
(x + y) \leq 1 - c_3 \exp[-\int [x f_C + y f_D] dt] \leq 1,
$$

$$(13)$$

where $c_3$ is the initial density dependent constant. Hence, we find that

$$
0 \leq x + y \leq 1,
$$

$$(14)$$

i.e., the overall species density $x + y$ eventually remains bounded within the region [0, 1]. This boundedness within the closed interval [0, 1] allows us to relate the possible emerging dynamics of the system (8) to physically implementable scenarios with biological relevance. When the sum of the population density $(x + y)$ is precisely one, the available free space is zero, as $z = 1 - (x + y)$. i.e., there is no reproductive opportunity accessible to any individual in that situation with $z = 0$. When $(x + y) = 0$, then individually $x = 0$ and $y = 0$. Hence, all individuals die, and $z$

is equal to one. Thus despite the presence of ample free space, all the individuals are extinct under that circumstances.

## 3.2 Coexistence of different stationary points depending on the initial conditions

Next, we point out the multistable dynamics of our model (8), resulting in the system's vulnerability to small perturbations. Initially, we set the parameter values at $\xi = 0.38$, $\beta = 1.1$, $\eta = 0.85$, $p = 0.30$, $\sigma_1 = 0.36$, $\sigma_2 = 0.25$ and $\mu = 0$ in Fig 1(a). We vary the initial conditions $(x_0, y_0)$ within the interval $[0, 1] \times [0, 1]$ maintaining the inequality $(x_0 + y_0) \in [0, 1]$. We find that the dynamics switching between two stationary points, viz. $E_0 = (0, 0)$ and $E_1 = (x^*, 0)$. We analytically calculate the stationary points of the system (8) in the absence of the mutation, i.e., with $\mu = 0$. We trace out four different stationary points,

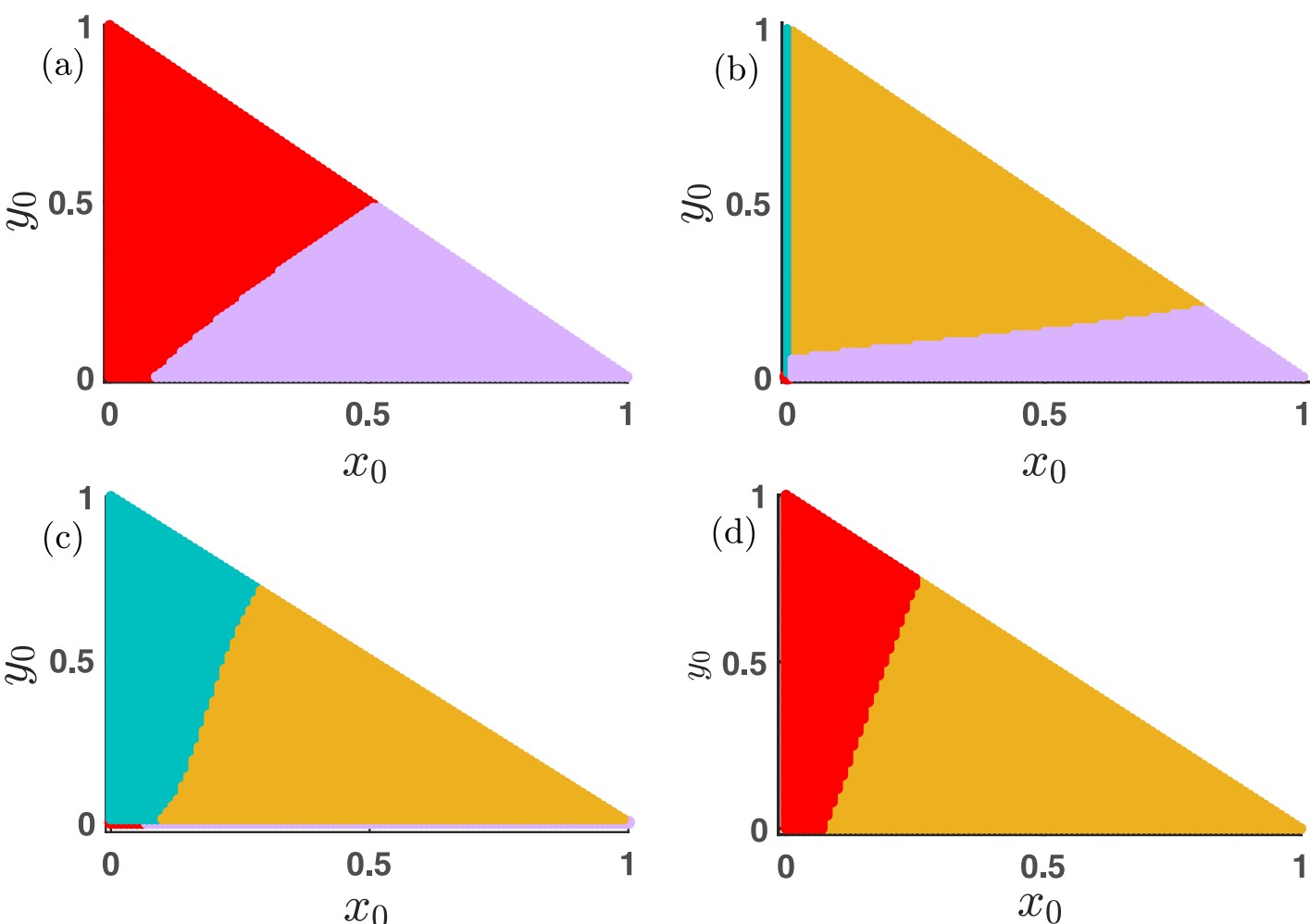

**Fig 1. Alternations between multiple co-existing steady states depending on initial conditions.** The coexistence of various stable states is portrayed here by varying the initial conditions maintaining $0 \leq (x_0 + y_0) \leq 1$. Red points signify the extinction equilibrium $E_0$. Violet points represent the defector-free steady state $E_1$. The cooperator-free stationary points $E_2$ are shown by sea blue points, and the coexistence equilibrium $E_3$ is plotted using yellow points. The mutation-free model in subfigures (a-c) allows four stable steady states to coexist. However, subfigure (d) supports only bistability for the chosen parameters' values. The parameter values for each of these subfigures are (a) $\xi = 0.38$, $\beta = 1.1$, $p = 0.30$, $\sigma_1 = 0.36$, $\sigma_2 = 0.25$, and $\mu = 0$. (b) $\xi = 0.15$, $\beta = 1.06$, $p = 0.69$, $\sigma_1 = 0.75$, $\sigma_2 = 0.25$, and $\mu = 0$. (c) $\xi = 0.15$, $\beta = 1.1$, $p = 0.10$, $\sigma_1 = 0.10$, $\sigma_2 = 0.25$, and $\mu = 0$. (d) $\xi = 0.30$, $p = 0.30$, $\beta = 1.1$, $\mu = 0.02$, $\sigma_1 = 0.30$, and $\sigma_2 = 0.20$. Other parameter is $\eta = 0.85$.

1. $E_0 = (0, 0)$ reveals all individuals die. This point is locally stable when

$$\begin{cases} \sigma_1 < \xi, \quad \text{and} \\ \sigma_2 < \xi. \end{cases}$$

2. $E_1 = (x^*, 0)$ exhibits a society free from any defectors. Here,

$$x^* = \frac{(1 - 2\sigma_1) \pm \sqrt{(1 - 2\sigma_1)^2 - 4(\xi - \sigma_1)(1 - \sigma_1)}}{2(1 - \sigma_1)}$$

and $x^* \in (0, 1]$. The stability criteria is given by

$$\begin{cases} \xi > (1 - \sigma_1)(2x^* - 3x^{*2}) + \sigma_1(1 - 2x^*) \quad \text{and} \\ \xi > (\beta - \sigma_2)(x^* - x^{*2}) + \sigma_2(1 - x^*). \end{cases}$$

3. $E_2 = (0, y^*)$ represents that we have only left with defectors. Here, $y^* = \frac{(2p\eta - \eta - 2\sigma_2)}{2(2p\eta - \eta - \sigma_2)} \pm$

$\frac{\sqrt{(2p\eta - \eta - 2\sigma_2)^2 - 4(2p\eta - \eta - \sigma_2)(\xi - \sigma_2)}}{2(2p\eta - \eta - \sigma_2)}$ and $y^* \in (0, 1]$. This stationary state is stable if

$$\begin{cases} \xi > \sigma_1(1 - y^*) - \sigma_1(y^* - y^{*2}) \quad \text{and} \\ \xi > \sigma_2(1 - 2y^*) + (2p\eta - \eta - \sigma_2)(2y^* - 3y^{*2}). \end{cases}$$

4. $E_3 = (x^{**}, y^{**})$ indicates the coexistence equilibrium offering the survival of cooperation and defection simultaneously. Here, $y^{**} = \frac{(1 - \beta - \sigma_1 + \sigma_2)x^{**} + \sigma_1 - \sigma_2}{2p\eta - \eta + \sigma_1 - \sigma_2}$, and $x^{**}$ satisfies

the equation, $(1 - \sigma_1)\tau x^{**} - \sigma_1 \left( \frac{(1 - \beta - \sigma_1 + \sigma_2)x^{**} + \sigma_1 - \sigma_2}{2p\eta - \eta + \sigma_1 - \sigma_2} \right) \tau + \sigma_1\tau - \xi = 0$, where,

$\tau = \frac{(2p\eta - \eta + \sigma_1 - \sigma_2)(1 - x^{**}) - (1 - \beta - \sigma_1 + \sigma_2)x^{**} + \sigma_2 - \sigma_1}{(2p\eta - \eta + \sigma_1 - \sigma_2)}$. $x^{**}$ and $y^{**}$ both should lie within the interval

$(0, 1)$. The local stability yields the conditions for the stability of $(x^*, y^*)$ are

$$\begin{cases} 2\xi > (\beta - \sigma_2)(x^{**} - x^{**2} - 2x^{**}y^{**}) \\ +(1 - \sigma_1)(2x^{**} - 3x^{**2} - 2x^{**}y^{**}) \\ -\sigma_1(y^{**} - y^{**2} - 2x^{**}y^{**}) \\ +(2p\eta - \eta - \sigma_2)(2y^{**} - 3y^{**2} - 2x^{**}y^{**}) \\ +\sigma_1(1 - 2x^{**} - y^{**}) + \sigma_2(1 - 2y^{**} - x^{**}), \quad \text{and} \\ \beta - \sigma_2)(x^{**} - x^{**2} - 2x^{**}y^{**}) \\ +(2p\eta - \eta - \sigma_2)(2y^{**} - 3y^{**2} - 2x^{**}y^{**}) \\ +\sigma_2(1 - x^{**} - 2y^{**}) - \xi][(1 - \sigma_1)(2x^{**} - 3x^{**2} \\ -2x^{**}y^{**}) - \sigma_1(y^{**} - y^{**2} - 2x^{**}y^{**}) \\ +\sigma_1(1 - 2x^{**} - y^{**}) - \xi] > [(\sigma_1 - 1)x^{**2} \\ -\sigma_1(x^{**} - x^{**2} - 2x^{**}y^{**}) - \sigma_1 x^{**}][(\beta - \sigma_2)(y^{**} \\ -y^{**2} - 2x^{**}y^{**}) - (2p\eta - \eta - \sigma_2)y^{**2} - \sigma_2 y^{**}]. \end{cases}$$

Clearly, the chosen parameters satisfy the local stability criteria of both stationary points $E_0$ and $E_1$. Note that $E_1$ leads to two different values for the selected values of the parameters, out of which (0.34751, 0) always remains locally stable, and (0.0899, 0) is unstable. Thus, we see the appearance of two different stationary points $E_0$ (shown by red points) and $E_1$ (displayed

by violet points) in the basin of attraction portrayed in Fig 1(a). Such toggling between alternate stable stationary points is one of the generic features in some biological systems involving the fundamental processes of life [63–65] and in a few nonlinear dynamical systems [66–69]. The initial condition $(x_0, y_0) = (0, 0)$ always helps the system (8) to converge to the stationary point $E_0$. The system is never able to give rise to the survival of any cooperators and defectors without the presence of any individual at the beginning. Since the chosen initial point $(0, 0)$ itself is the stationary state, the system always stabilizes in the $(0, 0)$ stationary point irrespective of the parameter values. Fig 1(b) is drawn with $\xi = 0.15$, $\beta = 1.06$, $\eta = 0.85$, $p = 0.69$, $\sigma_1 = 0.75$, $\sigma_2 = 0.25$, and $\mu = 0$. Similarly, we find the system (8) in the absence of mutation (i.e., $\mu = 0$) converges to $E_0$ for the single initial condition $(x_0, y_0) = (0, 0)$. It is anticipated that the choice of $x_0 = 0$ always leads to the cooperator-free steady state. The initial absence of cooperators in the mutation-free model will not entertain any exposure for the cooperators in the long run. The line of initial conditions $x_0 = 0$ and $y_0 \neq 0$ produces the stationary state $E_2$ (sea blue points in Fig 1(b)). Apart from these two steady states, the mutation-free system also switches between $E_1$ (violet points) and $E_3$ (yellow points) depending on the suitable choice of initial conditions (See Fig 1(b)). Thus, for the same choices of parameters' values, the system flips between four alternate steady states depending on the initial densities of cooperators and defectors. Similarly, we find the system (8) with the parameters' values $\xi = 0.15$, $\beta = 1.1$, $\eta = 0.85$, $p = 0.10$, $\sigma_1 = 0.10$, $\sigma_2 = 0.25$ and $\mu = 0$ converges to all these four stationary points. All population extincts for the initial conditions ranging from $(0, 0)$ to $(0.06, 0)$ (See red points in Fig 1(c)). The system (8) can be solved analytically with $\sigma_1 = \sigma_2 = \mu = 0$ in the absence of defectors (i.e., $y = 0$) as follows,

$$
x(t) = \begin{cases}
0, \\[4pt]
\dfrac{1}{2} \pm \dfrac{(1 - 4\xi)^{\frac{1}{2}}}{2}, \\[10pt]
c_4 - t = \dfrac{1}{2\xi} \left[ -\log(\xi + x(x - 1)) + 2\log x + \dfrac{2\tan^{-1}(\dfrac{2x - 1}{\sqrt{4\xi - 1}})}{\sqrt{4\xi - 1}} \right].
\end{cases}
$$

where $c_4$ is the integrating constant.

Similarly, the system (8) can be solved analytically with $\sigma_1 = \sigma_2 = \mu = 0$ in the absence of cooperators (i.e., $x = 0$) as follows

$$
y(t) = \begin{cases}
0, \\[4pt]
\dfrac{1}{2} \pm \dfrac{[-\eta(2p - 1)(\eta + 4\xi - 2\eta p)]^{\frac{1}{2}}}{2\eta(2p - 1)}, \\[10pt]
c_5 - t = \dfrac{1}{2\xi} \Bigg[ -\log(\xi + (2p - 1)\eta(y - 1)y) \\[10pt]
\qquad +2\log y + \dfrac{2\sqrt{(2p - 1)\eta}\tan^{-1}\dfrac{\sqrt{(2p - 1)\eta}(2y - 1)}{\sqrt{4\xi - (2p - 1)\eta}}}{\sqrt{4\xi - (2p - 1)\eta}} \Bigg]
\end{cases}
$$

where $c_5$ is the integrating constant.

Fig 1 is plotted by solving the differential Eq (8) by varying the initial conditions within $[0, 1] \times [0, 1]$ with fixed step-length $\delta x_0 = \delta y_0 = 0.01$ and maintaining $0 \leq x_0 + y_0 \leq 1$. To solve our proposed system (8) numerically, we use the 4th order Runge-Kutta (RK4) method with $20 \times 10^5$ iterations with fixed integration step length $\delta t = 0.01$. The final point $(x, y)$ is stored to

decide the asymptotic dynamics of the system. Other initial conditions with $y_0 = 0$ stabilize the dynamics in the defector-free state $E_1$ (violet points). We also track a fair portion of the basin in Fig 1(c), where the system (8) with $\mu = 0$ converges to either $E_2$ (sea blue points) or $E_3$ (yellow points) depending on the choice of initial conditions.

Although we choose the mutation-free model for the Fig 1(a)–1(c), we consider the contribution of $\mu$ in Fig 1(d). We generate Fig 1(d) with $\xi = 0.30$, $p = 0.30$, $\beta = 1.1$, $\eta = 0.85$, $\mu = 0.02$, $\sigma_1 = 0.30$, and $\sigma_2 = 0.20$. This non-zero $\mu$ leads to the disappearance of two stationary points, (i) the defector-free steady state $E_1$, and (ii) the cooperator-free steady state $E_2$. The symmetric mutation from one species to another species always gives them two feasible opportunities. Either both strategies survive or all the individuals perish for $\mu \neq 0$. We also mathematically derive two possible stationary points as follows,

1.  The extinction equilibrium $E_0 = (0, 0)$, which is stable under the conditions,

$$\begin{cases} 2(\xi + \mu) > \sigma_1 + \sigma_2, \\ \mu^2 < (\xi + \mu - \sigma_1)(\xi + \mu - \sigma_2). \end{cases}$$

2.  The interior equilibrium $E_3 = (x^*, y^*)$ becomes stable, if

$$\begin{cases} 2(\mu + \xi) > (1 - \sigma_1)(2x^* - 3x^{*2} - 2x^*y^*) \\ +(2p\eta - \eta - \sigma_2)(2y^* - 3y^{*2} - 2x^*y^*) \\ -\sigma_1(y^* - y^{*2} - 2x^*y^*) \\ +(\beta - \sigma_2)(x^* - x^{*2} - 2x^*y^*) \\ +\sigma_1(1 - 2x^* - y^*) + \sigma_2(1 - x^* - 2y^*), \\ \text{and,} \\ \sigma_1)(2x^* - 3x^{*2} - 2x^*y^*) \\ -\sigma_1(y^* - y^{*2} - 2x^*y^*) \\ +\sigma_1(1 - 2x^* - y^*) - \xi - \mu][(\beta - \sigma_2)(x^* - x^{*2} \\ -2x^*y^*) + (2p\eta - \eta - \sigma_2)(2y^* - 3y^{*2} \\ -2x^*y^*) + \sigma_2(1 - x^* - 2y^*) - \xi - \mu] > [(\sigma_1 - 1)x^{*2} \\ -\sigma_1(x^* - x^{*2} - 2x^*y^*) - \sigma_1 x^* + \mu][(\beta \\ -\sigma_2)(y^* - y^{*2} - 2x^*y^*) - (2p\eta - \eta \\ -\sigma_2)y^{*2} - \sigma_2 y^* + \mu], \\ \text{where } x^* \text{ and } y^* \text{ satisfy the equations :} \\ x^*[(1 - \sigma_1)x^*(1 - x^* - y^*) - \sigma_1 y^*(1 - x^* \\ -y^*) + \sigma_1(1 - x^* - y^*) - \mu - \xi] + \mu y^* = 0, \\ \text{and} \\ y^*[(\beta - \sigma_2)x^*(1 - x^* - y^*) \\ +(2p\eta - \eta - \sigma_2)y^*(1 - x^* - y^*) \\ +\sigma_2(1 - x^* - y^*) - \mu - \xi] + \mu x^* = 0. \end{cases}$$

The compelling evidence of bistability under the same choice of parameters' values is recognized in Fig 1(d).

## 3.3 Emergent dynamics in absence of mutation

We examine the impact of the parameters $p$, $\beta$, $\eta$, and $\xi$ on the system (8) with $\mu = 0$ in Fig 2. For the comparability, we choose $\sigma_1 = \sigma_2 = 0$ in subfigures (a), (c) and (e). Also, we set $\sigma_1 \neq 0$ and $\sigma_2 \neq 0$ for the subfigures (b), (d) and (f). In the absence of mutation ($\mu = 0$) and free space

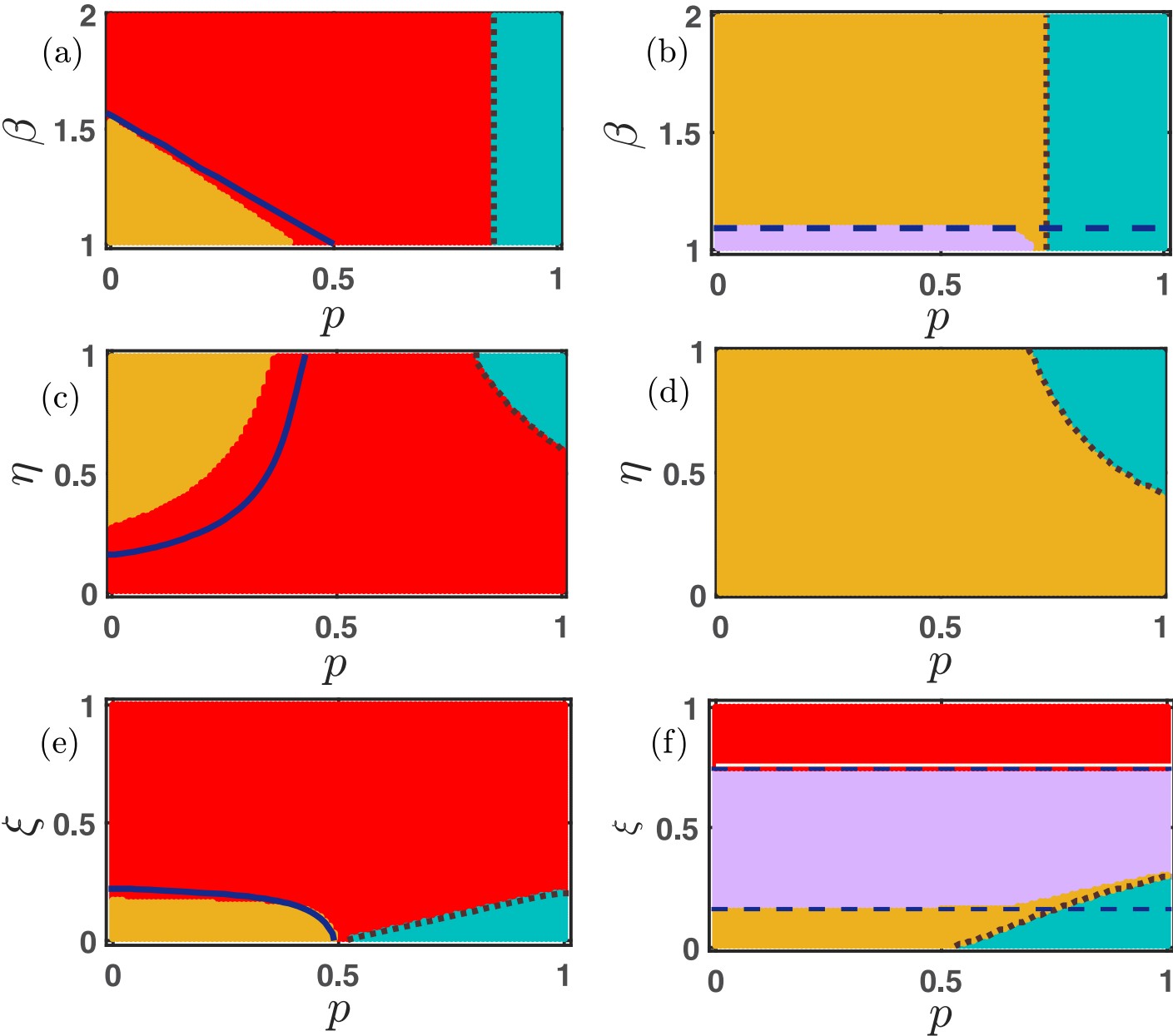

**Fig 2. The interplay of different parameters in the mutation-free model (8).** $p$ is the probability of playing the PD game. We investigate the role of $p$ in our proposed model (8) in the absence of the mutation. The figures (a), (c), and (e) of the left-handed column are drawn with $\sigma_1 = \sigma_2 = 0$. The remaining figures in the right-hand column are plotted with non-zero values of $\sigma_1$ and $\sigma_2$. The advantages provided by the free space help to promote cooperation, as portrayed through the broad area of yellow and violet regions in figures (b), (d), and (f). Red, sea blue, violet, and yellow depict the stable extinct state, cooperator-free state, defector-free state, and co-existence state, respectively. We also plot our calculated curves from the local stability analysis of the stationary points. The white line represents that of $E_0$, whereas the blue line portrays that of the coexistence equilibrium $E_3$. The brown dotted line is the stability curve for the cooperator-free stationary state $E_2$, while the blue dashed line represents that of the defector-free steady state $E_1$. The mismatches with the stability curve in a few places are due to the multistable behavior of our constructed model (8). Other parameter values are kept fixed at (a-d) $\xi = 0.15$, (c-f) $\beta = 1.1$, (a,b,e,f) $\eta = 0.85$, (b,d,f) $\sigma_1 = 0.75$, (b,d,f) $\sigma_2 = 0.25$ and $\mu = 0$.

induced benefits ($\sigma_1 = \sigma_2 = 0$), the defector-free $E_1 = \left( x^* = \frac{1 \pm \sqrt{1-4\xi}}{2}, 0 \right)$ is unable to stabilize.
Although $E_1$ exists for $0 < \xi \leq \frac{1}{4}$, but two eigenvalues of the Jacobian of the linearized system
are

$$\begin{cases} \lambda_1 = 2x^* - 3x^{*2} - \xi, \\ \lambda_2 = \beta x^* - \beta x^{*2} - \xi = \xi(\beta - 1). \end{cases}$$

Since $\beta > 1$ and $\xi > 0$, $\lambda_2$ is always positive. Consequently, $E_1$ is always unstable. This result
is physically meaningful, as both the chosen games (PD and SD) do not encourage a defection-
free society without any supporting mechanism. However, we find three different steady states
in Fig 2(a), 2(c) and 2(e). These stationary points and their corresponding stability analysis are
given below,

1.  The extinction equilibrium $E_0 = (0, 0)$ (red points in subfigures (a), (c) and (e) of Fig 2)
    always exists and is always locally stable.

2.  $E_2 = (0, y^{**})$ is the cooperator-free stationary point (sea blue points in subfigures (a), (c)
    and (e) of Fig 2), where $y^{**} = \frac{1}{2} \pm \frac{\sqrt{\eta^2(2p-1)^2 - 4\eta\xi(2p-1)}}{2\eta(2p-1)}$. This state exists under the condition
    $0 \leq 4\xi\eta(2p-1) \leq \eta^2(2p-1)^2$, and becomes stable if $\xi > \eta(2p-1)(2y^{**} - 3y^{**2})$.

3.  $E_3 = (x^*, y^*)$ allows the coexistence of both cooperators and defectors (yellow points in sub-
    figures (a), (c) and (e) of Fig 2), where $x^* = \frac{1 \pm \sqrt{1-4\kappa\xi}}{2\kappa}$, and $y^* = \frac{1-\beta}{2p\eta - \eta}x^*$, with $\kappa = \frac{1-\beta+2p\eta-\eta}{2p\eta-\eta}$.
    The stationary state is stable under the following conditions,

$$\begin{cases} 2\xi > (2+\beta)x^* - (3+\beta)x^{*2} - 2(\beta+1)x^*y^* \\ +(2p\eta - \eta)(2y^* - 3y^{*2} - 2x^*y^*), \\ (2x^* - 3x^{*2} - 2x^*y^* - \xi)(\beta x^* - \\ \beta x^{*2} - 2\beta x^*y^* + (2p\eta - \eta)(2y^* - 3y^{*2} - 2x^*y^*) \\ -\xi) + x^{*2}(\beta y^* - \beta y^{*2} - 2\beta x^*y^* - (2p\eta - \eta)y^{*2}) > 0. \end{cases}$$

With increasing $p$, players are prone to play the PD game. Hence, we can not anticipate the
survival of cooperators for large $p$. Thus, for large $p$, either the extinction equilibrium $E_0$ or the
cooperator-free stationary point $E_2$ stabilizes in Fig 2(a), 2(c) and 2(e). However, the free-space
induced benefits in Fig 2(b), 2(d) and 2(f) facilitate the emergence of cooperation and stabilize
the interior equilibrium $E_3$. Even, Fig 2(f) provides a range in the $p - \xi$ parameter space, where
the defector-free stationary point $E_1$ stabilizes. The comparative study between the left and
right column of Fig 2 ensures the encouraging role of free space in the promotion of coopera-
tion under suitable circumstances.

With increasing $\beta$, the defectors are getting extra aid, and thus, we get the stabilization of
the extinction equilibrium $E_0$ (red zone) in Fig 2(a). However, the presence of free space assis-
tance converts that area into a coexistence zone (yellow zone) in Fig 2(b). The increment of $p$
in both subfigures (a-b) permits only the existence of defectors, as people are playing mostly
the PD game in such circumstances. Defectors are getting a favorable atmosphere in the PD
game for our chosen PD game parameters' values. A similar sort of stabilization of cooperator-
free steady state (sea blue area) is observed in subfigures (c-d) for larger values of $p$. However,
lower values of $p$ provide the opportunity for playing the SD game. Hence, coexistence equilib-
rium (yellow region) stabilizes in Fig 2(c) and 2(d) for smaller $p$. Nevertheless, the inclusion of

free space induced benefits helps to broaden the region of coexistence in the $p - \eta$ parameter plane, as shown in Fig 2(d). The comparison between the Fig 2(e) and 2(f) suggests that appropriate non-zero values of $\sigma_1$ and $\sigma_2$ entertain the stabilization of a defector-free steady state (violet region). Hence, we can trace a fair portion of violet points in the $p - \xi$ parameter space of the subfigure (f). However, too large a mortality rate reduces the opportunity of survivability of any individual, resulting in the stabilization of the extinction equilibrium $E_0$. We focus on the effect of mortality rate $\xi$ more elaborately in Fig 3.

As expected, higher values of $\xi$ constantly enlarge the chances of extinction. Thus, we observe a fair portion of the red region in Fig 3. Nevertheless, the amount of this red area is considerably lesser in the right column of Fig 3 compared to the left column. We introduce the non-zero values of $\sigma_1$ and $\sigma_2$ in the right column of this figure. These free space induced benefits encourage maintaining a defector-free society, as shown in Fig 3(b) and 3(d). We choose

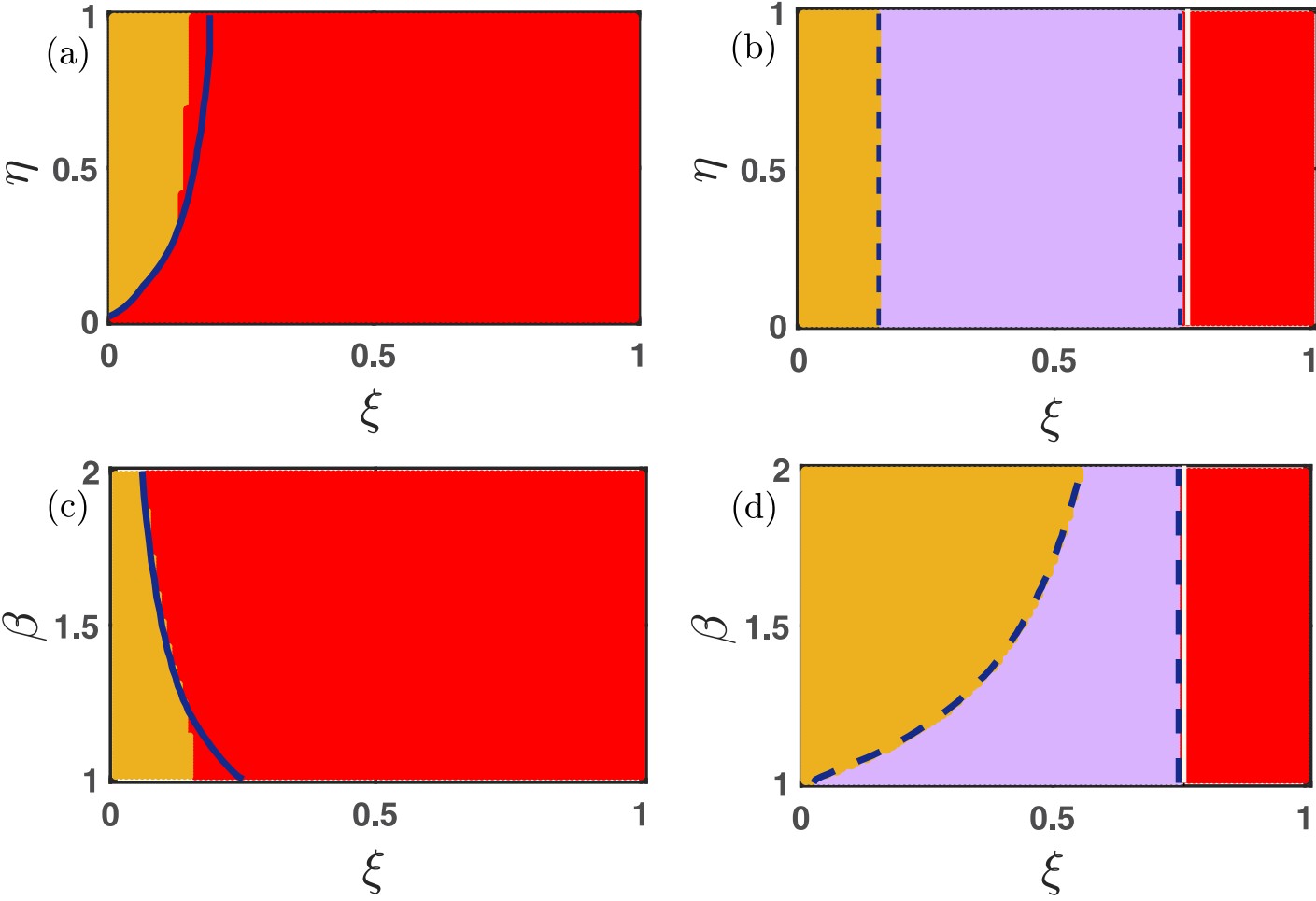

**Fig 3. Influence of death rate $\xi$ in the mutation-free model (8).** The influence of game parameters $\beta$ and $\eta$ and the mortality rate $\xi$ on the emergent dynamics of the model (8) in the absence of mutation is examined here. Subfigures (a) and (c) are drawn with $\sigma_1 = \sigma_2 = 0$. While subfigures (b) and (d) are generated with $\sigma_1 = 0.75$ and $\sigma_2 = 0.25$. Since, $\sigma_1 > \sigma_2$, we notice a fair portion of defector-free region (violet points) in subfigures (b) and (d). In the absence of free space induced benefits, we are unable to trace such defector-free regions in subfigures (a) and (c). The red region reflects the disappearance of all individuals for relatively high values of death rate $\xi$. Note that we keep the value of $p$ fixed at 0.3. Thus, the system gets more opportunities to play the SD game, which generally encourages stabilizing the coexistence equilibrium. Therefore, we notice lower values of $\xi$ will lead to the convergence towards the coexistence equilibrium $E_3$ (yellow points). $\beta$ is kept fixed at 1.1 for the subfigures (a-b) and $\eta$ is set at 0.85 for subfigures (c-d). We iterate the system (8) with $\mu = 0$ for $20 \times 10^5$ iterations with fixed integration step size $\delta t = 0.01$ and fixed initial condition $(x_0, y_0) = (0.35, 0.35)$.

$\sigma_1 = 0.75 > 0.25 = \sigma_2$ for Fig 3(b) and 3(d). i.e., the free space will provide an additional advantage for the cooperators, and thus depending on the other parameters, the defectors are vanished in the long run, as shown in Fig 3(b) and 3(d). In all of these subfigures of Fig 3, we trace a portion of yellow points depicting the survival of both cooperators as well as defectors simultaneously. We choose $p = 0.3$ in Fig 3. Thus, people get more chances to play the SD game, which facilitates the concurrence of both cooperation and defection. Thus, smaller values of $\xi$ provide an opportunity to coexist for all strategies, which is observed in Fig 3 with initial condition $(x_0, y_0) = (0.35, 0.35)$. We plot the stability curve of all stationary points in Figs 2–5 as follows,

(i) The brown dotted line for the cooperator-free steady state $E_2$,

(ii) the blue dashed line for the defector-free steady state $E_1$,

(iii) the solid blue line for the interior equilibrium $E_3$, and

(iv) the solid white line for the extinction equilibrium $E_0$.

In Fig 4, we fix the value of $\sigma_2$ at 0.25 and examine the role of $\sigma_1$. Since $\sigma_1$ represents the free space induced benefits towards the cooperators, thus enhancement of its ($\sigma_1$) value will help in

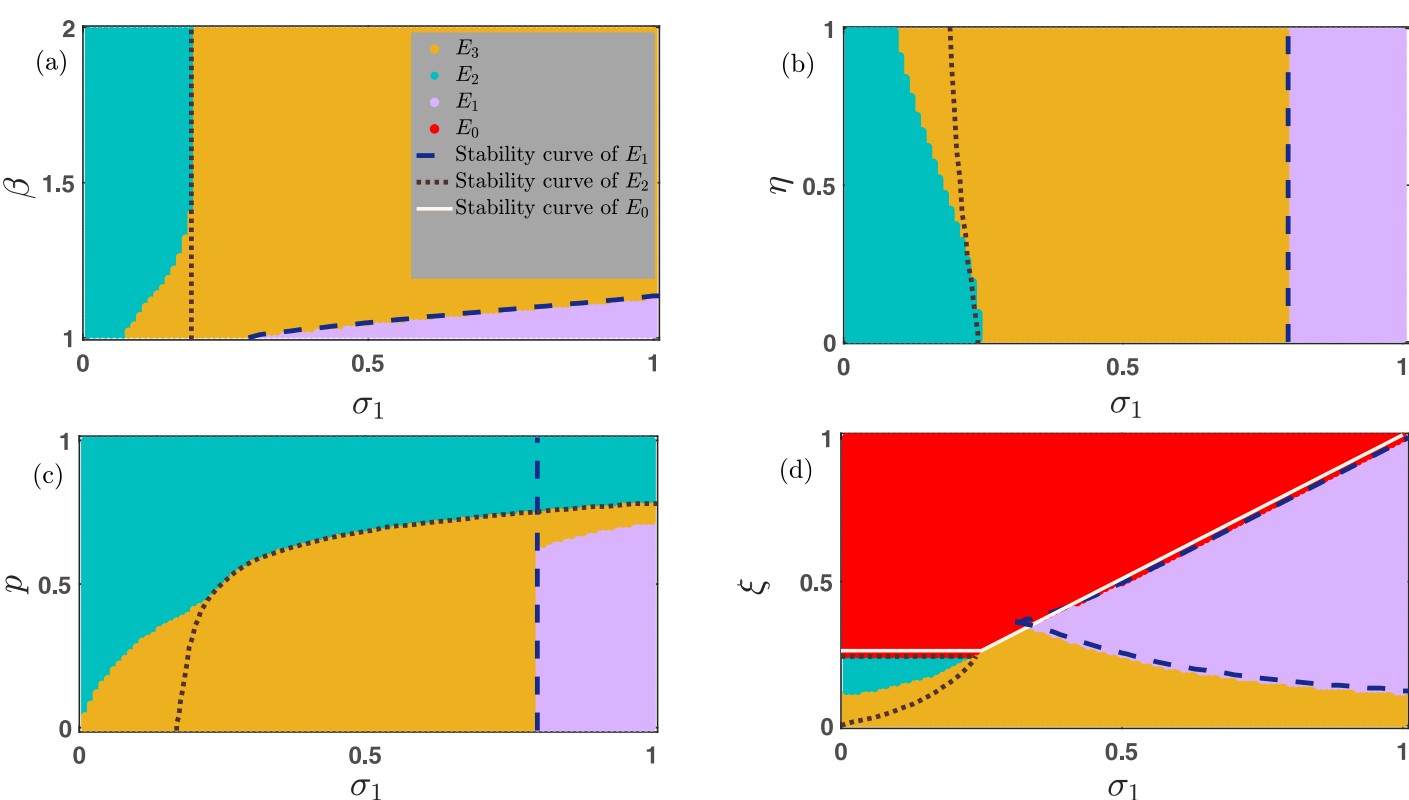

**Fig 4. Importance of $\sigma_1$ in the enhancement of cooperation.** Four different parameter spaces (a) $\sigma_1 - \beta$, (b) $\sigma_1 - \eta$, (c) $\sigma_1 - p$, and (d) $\sigma_1 - \xi$ are contemplated here with fixed initial condition $x_0 = 0.35$ and $y_0 = 0.35$. $\sigma_1$ is varied within $[0, 1]$, and the other parameters' values are for the subfigures (a) $\xi = 0.15$, $\eta = 0.85$, $p = 0.30$, $\sigma_2 = 0.25$, and $\mu = 0$. $\beta$ is varied within the open interval $(1, 2)$, (b) $\xi = 0.15$, $\beta = 1.1$, $p = 0.30$, $\sigma_2 = 0.25$ and $\mu = 0$. $\eta$ is varied within $(0, 1)$, (c) $\xi = 0.15$, $\beta = 1.1$, $\eta = 0.85$, $\sigma_2 = 0.25$, and $\mu = 0$. $p$ is varied within the closed interval $[0, 1]$, and (d) $\beta = 1.1$, $\eta = 0.85$, $p = 0.30$, $\sigma_2 = 0.25$ and $\mu = 0$. $\xi$ is varied within the interval $(0,1]$. The color code represents the following: (i) red represents the extinct state, (ii) yellow portrays the co-existence state, (iii) violet displays the defector-free state, and (iv) sea blue depicts cooperator-free state, respectively. We have run the numerical simulations for $20 \times 10^5$ iterations for each point and store the final value for determining the final asymptotic state. Increment of $\sigma_1$ contributes more to the cooperators' payoff, and hence, we observe a defector-free region for higher values of $\sigma_1$ depending on the other parameters. We draw the stability curves for $E_0$ (the solid white line), $E_1$ (the blue dashed line), $E_2$ (the brown dotted line), and $E_3$ (the solid blue line).

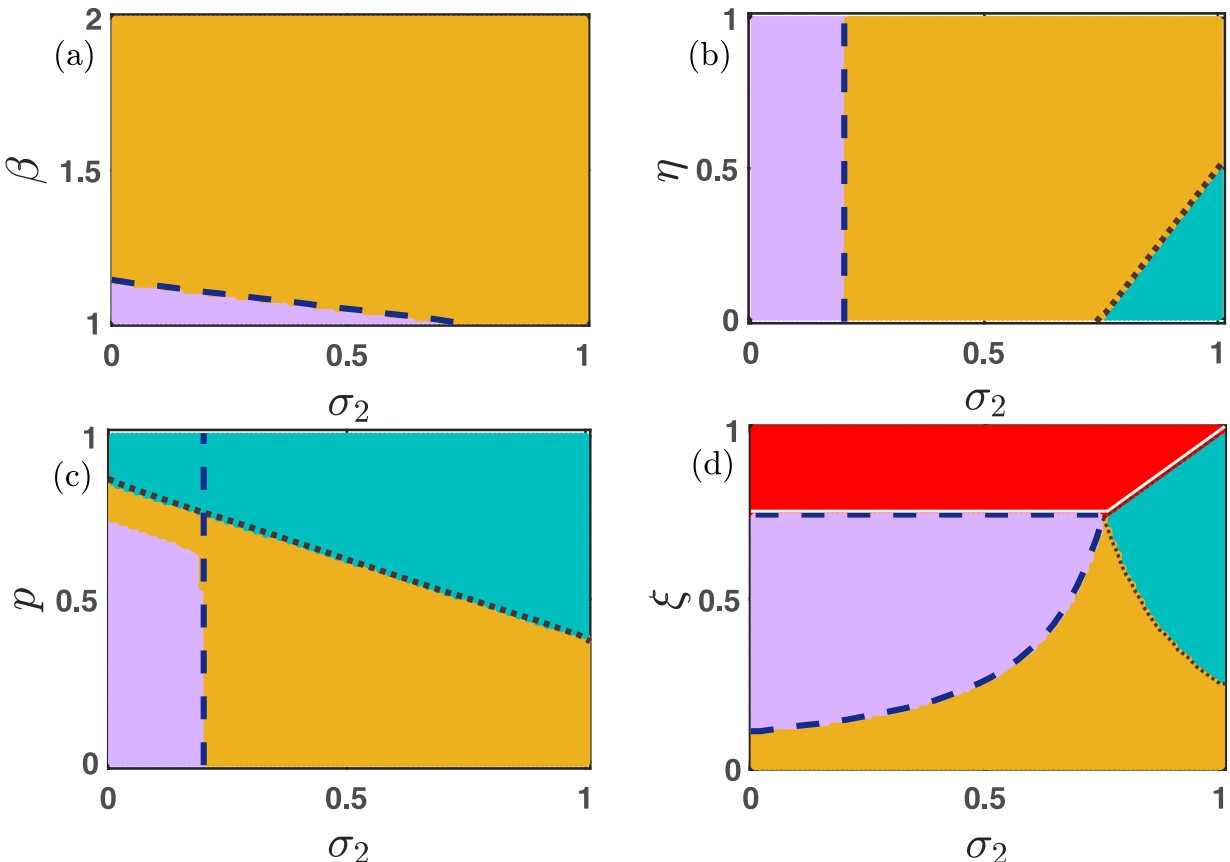

**Fig 5. Investigating the impact of $\sigma_2$ on the evolution of cooperation.** For the smaller values of $\sigma_2$, the mutation-free system gives rise to the defector-free (violet region) society. While larger values of $\sigma_2$ encourage the defectors and stabilize the cooperator-free stationary point $E_2$ (sea blue region). Suitable choices of parameters may favor the co-existence of cooperators as well as defectors. Even for a higher mortality rate $\xi$'s values, all individuals die in the red portion of the parameter space $\sigma_2 - \xi$ in subfigure (d). All the parameters are set at $\xi = 0.15$, $\eta = 0.85$, $\beta = 1.1$, $p = 0.30$, $\sigma_1 = 0.75$, and $\mu = 0$, unless it is varied. The simulations are computed for $20 \times 10^5$ iterations with fixed initial condition $(x_0, y_0) = (0.35, 0.35)$. The color codes are as follows: (i) red represents the extinct state $E_0$, (ii) yellow represents the co-existence state $E_3$, (iii) violet represents the defector-free state $E_1$, and (iv) sea blue represents cooperator free state $E_2$, respectively. We plot the stability curves corresponding to each stationary point in each subfigure.

promoting cooperation in the society. Fig 4 reflects the same scenario. A defector-free society (violet region) is noticed in all these subfigures. The increment of temptation parameter $\beta$'s value challenges the prevalence of cooperation. Thus, for a small $\sigma_1$, we find a defector dominated society (the sea blue region in Fig 4(a)). $\sigma_1$ proves to be a cooperator facilitating parameter as we detect a wide range of yellow regions in Fig 4(a), where both cooperators and defectors can coexist. Similarly, irrespective of the choice of $\eta$ in Fig 4(b), higher values of $\sigma_1$ provide all cooperators an extra benefit for survival, and thus, the stationary point $E_1$ (violet points) stabilizes. For the intermediate choice of $\sigma_1$, the stationary point $E_3$ (yellow) yields the stable coexistence of both strategies. However, cooperators strive to keep in existence for the lower values of $\sigma_1$, and we find the sea blue region of cooperator-free steady state. The parameter $p$ indicates the probability of playing the PD game. Thus, $p \to 1-$ always gives defectors a more favorable environment to survive. That's why we spot a sea blue portion in Fig 4(c) for higher values of $p$ and $\sigma_1$. When $p$ is small, people are prone to play the SD game; and thus, we get the coexistence of both strategies (the yellow region in Fig 4(c)) for smaller $p$ and $\sigma_1$. Nevertheless, a larger value of $\sigma_1$ with a moderate value of $p$ always provides a reasonable scope for

the cooperators to survive, and we discover a healthy portion of the stationary point $E_1$ (the violet region) in Fig 4(c). A higher mortality rate $\xi$ always results in the extinction of both cooperators and defectors, and we locate a huge red region in the $\sigma_1 - \xi$ parameter plane of Fig 4(d). We find a very tiny sea blue region in Fig 4(d), where defectors are only able to survive. But, as expected, a higher value of $\sigma_1$ always promotes the cooperation strategy, and we obtain a violet zone of defector-free stationary point and a yellow region of interior equilibrium $E_3$ in Fig 4(d).

Fig 5(a) shows that except for a smaller portion of the defector-free region (violet zone), the whole $\sigma_2 - \beta$ parameter space produces the coexistence of both cooperators and defectors depending on other parameters' values. Despite the increment of $\sigma_2$ and $\beta$, the cooperators are able to survive along with the defectors due to our choice of other parameters' values. The $\sigma_2 - \eta$ parameter space portrays that smaller values of $\sigma_2$ can not provide any benefit to the defectors, and stabilize the defector-free stationary point $E_1$ (violet region) irrespective of $\eta$'s value. However, if $\sigma_2$ increases, it will yield a window of opportunity for the defectors to thrive. We observe a wide coexistence region (yellow region) and a small area of cooperator-free stationary point (sea blue region)in Fig 5(b). $\sigma_2$ indicates the free-space induced benefits towards the defectors. So, it is expected that larger values of $\sigma_2$ always enhance the chances of defectors' survivability. Thus, we notice a cooperator-free region (sea blue zone) in both Fig 5(c) and 5(d). Nevertheless, larger values of $p$ enhance the chances of playing the PD game, where defectors get a favorable environment to survive. Thus, the cooperator-free sea blue region is found in Fig 5(c). We are also able to detect a small violet region of a defector-free environment in the $\sigma_2 - p$ parameter space. However, we trace a healthy portion of coexistence (yellow) too in Fig 5(c) due to our chosen parameters' values.

## 3.4 The influence of bidirectional mutation

Now we investigate the influence of bidirectional mutation on the long-term behavior of the nonlinear differential Eq (8). Since every species can mutate into the other at a specific uniform rate $\mu \in (0, 1]$, thus cooperators and defectors can not remain alive alone. Either all populations will die; otherwise, the dynamics will lead to the coexistence of all two species. Fig 4 already portrays the influence of free space-induced benefits on the cooperators in the absence of mutation. We scrutinize the impact of $\sigma_1$ under the influence of 50% mutation (i.e., $\mu = 0.5$) in Fig 6. As $\sigma_1$ increases, the cooperators are getting a better environment for survival. We find a portion of stable coexistence equilibrium in each two-dimensional parameter space for large $\sigma_1$ in Fig 6. Thanks to the mutation, the cooperators can not live alone. The white line in Fig 6 is the stability curve corresponding to the extinction equilibrium. The local stability analysis fits almost exactly with the numerical simulations in Fig 6 with fixed initial condition $(x_0, y_0) = (0.35, 0.35)$. There are a few places where the stability analysis fails to predict the stabilization of the extinction stationary point $(0, 0)$. This is mainly due to the multistable behavior of the proposed model 8.

As discussed, the increasing values of the parameters $\beta$, $\eta$, and $p$ always provide the defectors a favorable environment to dominate the cooperators. However, beyond a critical value of $\sigma_1$, both the cooperators and defectors can coexist, as shown in Fig 6(a)–6(c). The larger values of $\xi$ always hinder the evolution of cooperators as well as of defectors. However, an intermediate choice of $\sigma_1 - \xi$ favors the successful evolution of both strategies, as portrayed through Fig 6(d). The same feature is also noticeable in Fig 7. The complex evolutionary dynamics switch between two stationary points depending on the choices of parameters' values in Fig 7. For smaller values of $\sigma_2$, the defectors are not getting enough advantages to survive, and thus the extinction equilibrium $(0, 0)$ stabilizes in the red region of all subfigures of Fig 7. The choice of

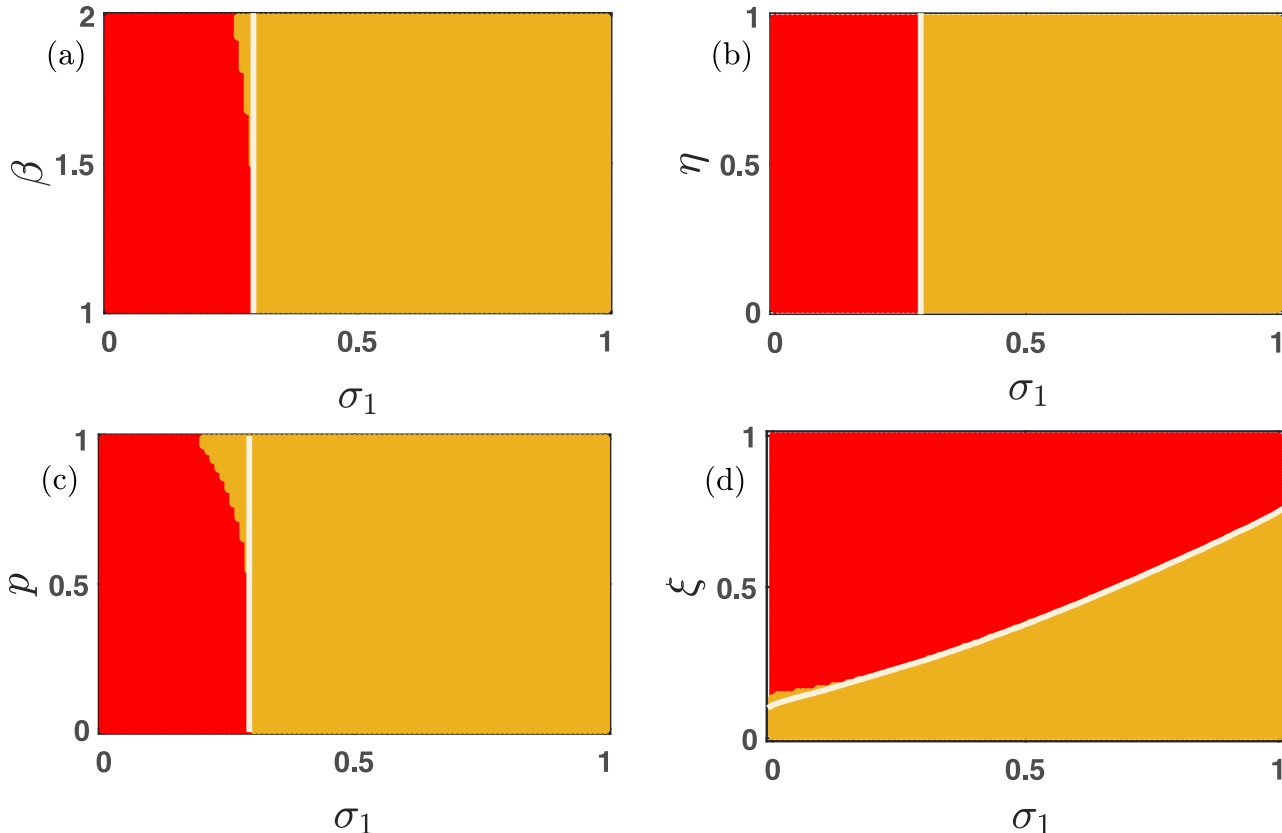

**Fig 6. The effect of altruist free space on the nonlinear dynamics of multigames with mutation.** The influence of free space-induced benefits by varying $\sigma_1$ within the closed interval [0, 1] is established. The whole population goes extinct in the red region, and the yellow area reflects the system's stable interior point, corresponding to the coexistence of all two strategies. All the parameters are kept fixed at $\xi = 0.25$, $\eta = 0.85$, $p = 0.30$, $\sigma_2 = 0.20$, $\beta = 1.1$, $\mu = 0.5$, unless it is varied. The initial condition is kept fixed at (0.35, 0.35). A notable difference is observed from (Fig 4), which is drawn in the absence of mutation. Larger values of $\sigma_1$ always facilitate the maintenance of cooperation, and the bidirectional mutation reinforces the system's inherent tendency to flow from cooperators to defectors. The solid white line in each figure is the analytically computed stability curve corresponding to the extinction equilibrium.

other parameters' values is also crucial for obtaining these stationary states. The parameter $\sigma_2$ benefits the defectors; hence the defectors can survive beyond a certain threshold of $\sigma_2$. The employed 50% mutation rate helps to flow a certain fraction of defectors into cooperators, and we have a moderate portion of coexistence state (yellow region) in Fig 7. The observed results may vary for different choices of initial conditions, as the system is multistable. We plot the boundary separating solid white lines in all subfigures by analyzing the local stability analysis of the extinction equilibrium. Clearly, this stability curve agrees well with our numerical simulations, and the places, where they don't agree with the numerical simulations, is due to the multistable behavior of our proposed model 8. All the simulations are done by iterating for $20 \times 10^5$ times with fixed integrating step length $\delta t = 0.01$. The last point is gathered to finalize the asymptotic state. All the codes to generate these figures are publicly available at Ref. [70].

Fig 8 demonstrates the importance of the parameters $\sigma_1$, $\sigma_2$ and $\xi$ for the enhancement of cooperation under the presence of mutation. The larger values of $\sigma_1$ and $\sigma_2$ facilitate the evolution of at least one species, and the positive mutation rate $\mu \in (0, 1]$ assures that species should mutate into the other. This mechanism will lead to the species' coexistence in a major portion (yellow region) of the two-dimensional parameter spaces represented in Fig 8(a) and 8(b). The results are further validated by plotting the stability curve (white solid lines) below which the

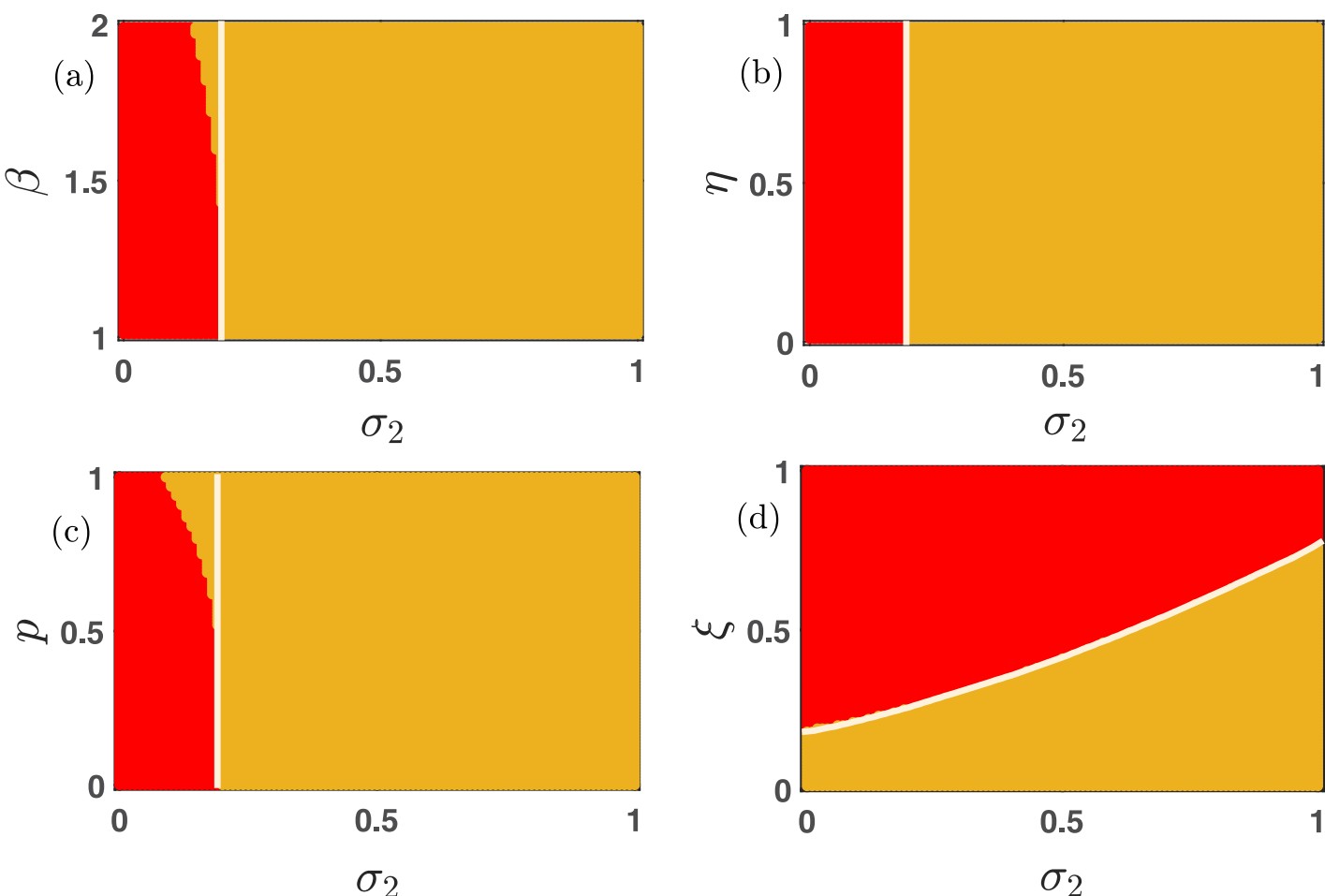

**Fig 7. The impact of free space-induced benefits on the defectors in the presence of the mutation.** Either all populations die (red region), or the symmetric mutation preserves the coexistence state (yellow area) in all these subfigures. Beyond a critical value of $\sigma_2$, both strategies mutate one into another, providing the opportunities for the coexistence of both species. The solid white line is the analytically derived stability curve to stabilize the extinction equilibrium. All the parameters are kept fixed at $\xi = 0.25$, $\eta = 0.85$, $p = 0.30$, $\sigma_1 = 0.30$, $\mu = 0.5$, $\beta = 1.1$, unless they are varied. The initial condition is kept fixed at $(0.35, 0.35)$. The slight mismatch of the stability of the extinction equilibrium $(0, 0)$ in subfigures (a) and (c) is due to the multistable dynamics of our proposed model (8).

stationary point $(0, 0)$ is locally stable. A higher mortality rate never entertains the evolution of both strategies; thus, we obtain the red region in Fig 8(c). This red region indicates the extinction equilibrium $(0, 0)$. Once again, we plot the stability curve (solid white line) in Fig 8(c), above which both the species should be extinct as per our local stability analysis. All the subfigures are drawn with fixed initial condition $(0.35, 0.35)$.

We inspect the influence of different parameters on the constructed model (8) in Fig 9. We keep fixed all parameters' values at $\xi = 0.25$, $\beta = 1.1$, $\mu = 0.5$, $p = 0.65$, $\eta = 0.85$, $\sigma_1 = 0.75$ and $\sigma_2 = 0.25$, unless they are varied. Since free space provides additional advantage to the cooperators compared to the defectors as we choose $\sigma_1 = 0.75 > \sigma_2 = 0.25$, we have a defector-free society at least initially with $\mu = 0$ in Fig 9(a). However, whenever the mutation rate $\mu$ becomes positive, each strategy can mutate into other. In this way, the density of the cooperators (red line) decreases, and the defectors' density (blue line) increases. Eventually, both densities almost become identical for $\mu \rightarrow 1-$. In of Fig 9(b), the vital role of $p$ is investigated in the one-dimensional bifurcation diagram. As $p \rightarrow 1-$, the defectors are getting the upper hand over the cooperators as $p$ indicates the probability of playing the PD game. PD game always provides

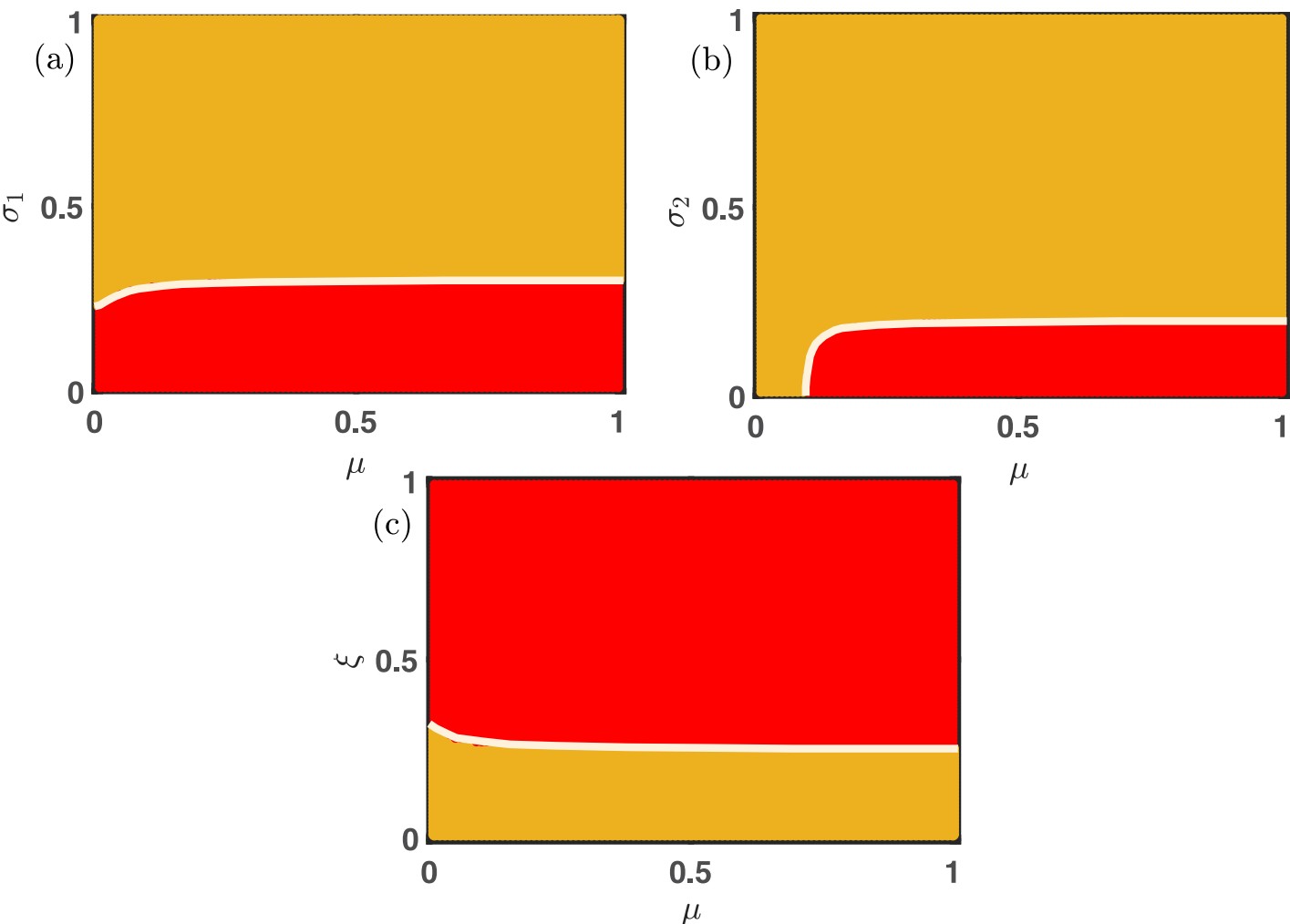

**Fig 8. Effect of bidirectional mutation on the emergent dynamics.** The positive mutation rate $\mu \in (0, 1]$ allows the system (8) to settle into two stationary points depending on the choice of other parameters' values. When the free space-induced benefits are small, all species are extinct, as depicted through subfigures (a-b). However, all strategies can coexist for suitable choices of other parameters. The red region reflects the extinction equilibrium, and the yellow region indicates the stable coexistence of cooperators and defectors. The solid white line represents the analytically derived stability curve below which the extinction equilibrium (0, 0) is locally stable in subfigures (a-b). The initial condition is chosen here as $(x_0, y_0) = (0.35, 0.35)$. Other parameters' values are kept fixed at $\xi = 0.25, \beta = 1.1, \eta = 0.85, p = 0.30, \sigma_2 = 0.20, \sigma_1 = 0.30$, unless they are varied. The moderate choice of mortality rate $\xi \in (0, 1]$ allows the species' coexistence in subfigure (c). Beyond a critical value of $\xi$, both the cooperators as well as the defectors die, as reflected through the red region of subfigure (c).

additional assistance to the defectors. All these results obtained in Fig 9 are consistent with the social dilemmas considered for constructing the model (8). Fig 9(b) points out that lower values of $p$ are always better for the maintenance of cooperation as small values of $p$ indicate more rational people are playing the SD game and the SD game always favors the coexistence of both strategies. Increasing the temptation parameter $\beta$ always uplifts the defectors' fraction $y$ (blue line). Thus, one needs to choose the value of $\beta$ wisely so that we obtain a moderate range of $\beta$ in Fig 9(c) allowing the coexistence of both strategies. When $\beta$ is small, we found the cooperators' density $x$ (red line) dominates the defectors' fraction $y$ (blue line). However for $\beta > 1.45$, $y$ is larger than $x$. Note that the results may alter for a different choice of initial condition as the system 8 is multistable. We plot Fig 9 with fixed initial condition (0.35, 0.35) and the code to generate this figure is freely available at [70]. Increasing the parameter $\eta \in (0, 1)$ can provide extra benefits to the defectors. Thus, the rate of increment of $y$ is slightly better

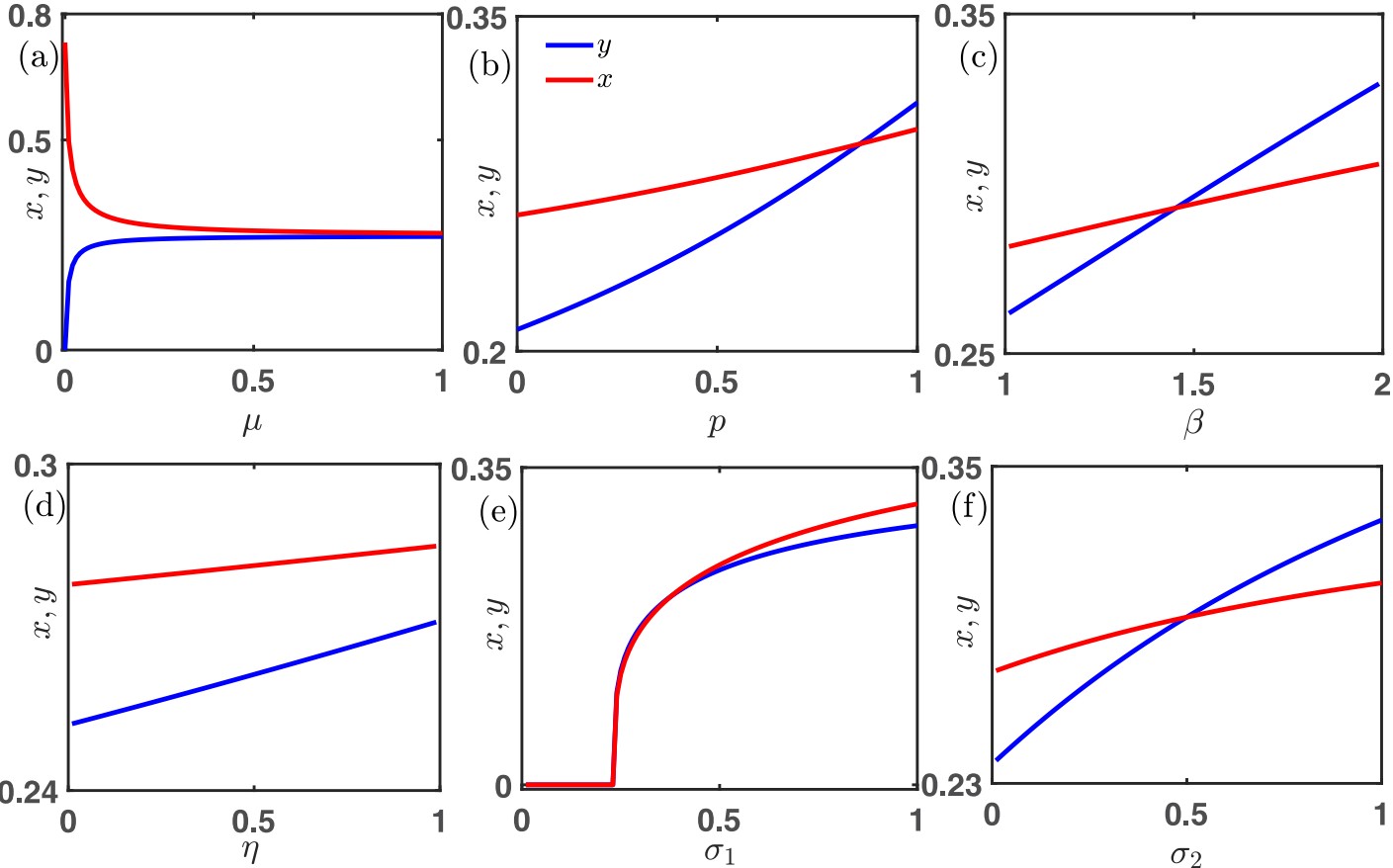

**Fig 9. Effects of various parameters in spreading of cooperative behavior.** Subfigure (a) reveals two contrasting scenarios, where the cooperators decrease with the increment of the mutation rate $\mu$. However, the defectors' fraction gradually increases with increasing $\mu$. In the absence of mutation (i.e., for $\mu = 0$), all defectors are extinct as the parameters values chosen to produce this subfigure is $\xi = 0.25$, $\beta = 1.1$, $\eta = 0.85$, $p = 0.65$, $\sigma_1 = 0.75$ and $\sigma_2 = 0.25$. Thus, the free space induced benefit towards cooperator $\sigma_1$ is greater than that of defector $\sigma_2$. Subfigure (b) shows that both species' densities gradually increase with the probability $p$ of playing the PD game. As the probability of playing the PD game increases, the increment of the fraction of defectors (blue line) gets over in margin with respect to the fraction of the cooperators (red line) after $p = 0.85$ depending on other parameters' values. This fact is obvious from the view that a higher possibility of playing the PD game gets a more beneficial ambiance for defectors. Subfigure (c) indicates the same phenomena as the value of the temptation payoff parameter $\beta$ increases, both the fractions escalate. Whereas at $\beta = 1.45$, both these fractions get the same value. After that value of $\beta$, the value of the fraction of defector (blue line) becomes more than that of the cooperators (red line). Subfigure (d) also highlights the increment of both the fractions as the punishment payoff parameter $\eta$ increases. But up to the highest received payoff value of $\eta$, the value of the fraction of defectors can never exceed the cooperators' fraction because the free space provides more benefit to the cooperators than the defectors. Subfigure (e) demonstrates that when $\sigma_1 \in [0, 0.25]$, both these fractions are extinct, as the mortality rate $\xi$ has more value than these free space-induced benefits. But as the value of $\sigma_1$ increases, the cooperator gets more advantages than defectors. Subfigure (f) shows that both the fractions increase with the enhancement of $\sigma_2$. Up to $\sigma_2 = 0.5$, the cooperators' density acquires more value than $y$, as the value of $\sigma_1 = 0.75$ is taken sufficiently high. After $\sigma_2 = 0.5$, $y$ becomes higher than $x$. All the subfigures are drawn by iterating the system (8) with $20 \times 10^5$ times with fixed integrating step length $\delta t = 0.01$ and fixed initial condition $(x_0, y_0) = (0.35, 0.35)$. The last point at $20 \times 10^5$ iteration is collected to identify the asymptotic state of the proposed model. Other parameters are kept fixed at $\xi = 0.25$, $\beta = 1.1$, $\mu = 0.5$, $p = 0.65$, $\eta = 0.85$, $\sigma_1 = 0.75$ and $\sigma_2 = 0.25$, unless they are varied.

than that of $x$ in subfigure (d) of Fig 9. However, other parameters' values are also crucial for stabilizing the competitive communities. Hence, $y$ (blue line) remains always lower than $x$ (red line) in Fig 9(d) for our chosen parameters' values.

Since the mutation rate $\mu$ is 50%, we can obtain only two stationary points of the model (8). Thus, initially, for a smaller choice of $\sigma_1$, extinction of both species prevails in Fig 9(e). Nevertheless, with an increment of $\sigma_1$, cooperators are getting further assistance from free space, and hence $x$ becomes positive beyond $\sigma_1 = 0.25$. Since $\mu = 0.5$, 50% of these cooperators mutate bidirectionally into the defectors, and thus, we have positive $y$ too in that range of $\sigma_1$. Since $\sigma_2 = \xi = 0.25$, thus we get the stabilization of $(0, 0)$ within the interval $\sigma_1 \in [0, 0.25]$. In fact, the

cooperators' density $x$ is slightly better than that of $y$ for the higher values of $\sigma_1$ in Fig 9(e). The choice of the initial condition and other parameters' values are vital in obtaining all these results. Similar observation can be found in Fig 9(f), where we examine the role of $\sigma_2$ in our proposed model (8). With increasing $\sigma_2$, the defectors will be benefited from free space, and hence, $y$ (blue line) dominates $x$ (red line) for larger values of $\sigma_2$. But since $\sigma_1$ is taken sufficiently large in this Fig 9(f), we initially have a small portion for smaller values of $\sigma_2$ where $x > y$. Fig 9 discloses suitable choices of all parameters' values not only entertain the coexistence of both strategies but also may promote the evolution of cooperation.

## 4 Concluding remarks

Our mutation-induced model provides widespread coexistence of both strategies under suitable choices of parameters' values. Such stable persistence of ecological communities strengthens the theory of concurrency. Our thorough analysis with several numerical simulations enhances our understanding of the mechanisms that drive the survival of cooperative behavior in the multigames consisting of both the Prisoner's Dilemma and the Snowdrift games. The inclusion of altruistic behavior of free space along with the mutation in our proposed evolutionary model seems to be a natural course of action as observed in many realistic settings. Our findings clearly demonstrate that there exists an optimal probability of playing each game so that people are more likely to cooperate in such circumstances. Throughout the study, we have pointed out the positive impact of diverse factors (parameters) on significant improvement of cooperation. We have shown that the selfless contribution of free space promotes the coexistence of all strategies efficiently, even in the absence of mutation.

In summary, our research indicates the proposed model (8) may possess four different stationary points in the absence of mutation. The exciting feature of this model is that the system never allows settling into a cooperator-free state in the lack of free space-induced benefits and mutation rate. This precise result is consistent with the chosen games as both the PD and SD games never encourage a cooperator-free society in the usual scenario. The merging of two games with different outcomes provides a more realistic representation of the concept of opinion formation. However, the numerical results presented here are highly sensitive to the variation of initial conditions. The positive invariance and boundedness of the model are analyzed too in this article. We have shown the viable choice of bidirectional mutation allows the system to switch between only two stationary states. Either all people will die, or both the strategies coexist in the eco-evolutionary model with mutation. This is an interesting angle of our research as our model brings forth stable biodiversity in the form of a heterogeneous population (mixed cooperator-defector state). Note that we have only focused on the equilibria of the corresponding dynamical system (8) throughout the article so that we can relate those stationary states from the game-theoretical point of view. Our simple model with mutation sheds light on how cooperation emerges in a complex society. The presented insightful results attest that altruistic behavior and mutation are advantageous for the spontaneous maintenance of biodiversity. We believe the presented results may help us understand the mechanism behind the coexistence of competing species through the co-evolution of both strategies.

## 5 Discussion: Limitation & future perspectives

Lastly, we discuss challenges related to our work that merits further investigation. Even though it is almost impossible to incorporate all complex, realistic relationships among social creatures using minimal modeling, it is nevertheless absolutely essential to investigate such models using elementary mathematical principles. In fact, the literature already provides a number of excellent models that offer a stimulating starting point for exploring many practical situations.

Chen et al. [71] proposed an elegant model aimed at understanding the co-evolutionary outcome of the strategies of the traffic management department, drivers, and pedestrians. Reference [72] proposes a hybrid machine learning model to predict traffic accidents. A novel autonomous system without any stationary points may offer hidden attractors, as revealed in Ref. [73]. It is almost impossible to incorporate even a partial list of relevant references to emphasize the importance of minimal modeling here. Motivated by all these works, we present a new model that can offer several valuable insights into the evolution of cooperation and altruism. However, our model is far from perfect. Instead of considering two-person interactions, the introduction of group interactions may yield a deeper understanding of decision-making. Interdependent networks may be an excellent choice to study the evolution of cooperation, as reflected through Refs. [20, 31, 62, 74]. This remains a promising future research generalization, including interdependent networks which may yield several complex dynamical behaviors other than stable equilibrium states.

One should observe that our approach works absolutely fine with other social dilemmas. For any two two-person games of the form $\begin{pmatrix} R & S \\ T & P \end{pmatrix}$, the system constructed with the help of our policy always remains bounded within the closed interval [0, 1]. However, the stationary points alter due to the change in the mathematical model. We have verified this boundedness by considering the snowdrift and the harmony game [75, 76] (results are not shown here). In fact, on the generality of our results, we can comment that a similar impact of mutation and free space can also be expected for other two-person games. It will be interesting to examine how the results may vary with other games like public goods games, rock paper scissor, etc. Besides introducing other realistic scenarios like apology and forgiveness [77], intention recognition [78], delay [79] etc., one may advance our understanding of the roots of cooperation in social and biological systems. Moreover, the impact of higher-order interactions [80] among the agents on the coevolution [81] of cooperation and synchronization [82–84] in a coupled network remains largely unexplored. The interdisciplinary researchers of complex systems can pay attention to this exciting topic for future research.

## Acknowledgments

The authors gratefully acknowledge the anonymous referees for their careful reading, insightful suggestions, and valuable remarks that helped considerably improve and readability of the manuscript. S.R. wants to convey his sincere gratitude to Gourab Kumar Sar, Md Sayeed Anwar, Srilena Kundu, and Subrata Ghosh of the Indian Statistical Institute, Kolkata, for their helpful discussion and suggestions.

## Author Contributions

**Conceptualization:** Sayantan Nag Chowdhury.

**Data curation:** Sourav Roy.

**Formal analysis:** Sourav Roy, Sayantan Nag Chowdhury.

**Funding acquisition:** Matjaž Perc.

**Investigation:** Sayantan Nag Chowdhury.

**Methodology:** Sourav Roy, Sayantan Nag Chowdhury.

**Project administration:** Sayantan Nag Chowdhury, Dibakar Ghosh.

**Resources:** Sayantan Nag Chowdhury.

**Software:** Sourav Roy.

**Supervision:** Sayantan Nag Chowdhury, Dibakar Ghosh.

**Validation:** Sourav Roy, Prakash Chandra Mali, Matjaž Perc, Dibakar Ghosh.

**Visualization:** Matjaž Perc, Dibakar Ghosh.

**Writing – original draft:** Sayantan Nag Chowdhury.

**Writing – review & editing:** Sourav Roy, Prakash Chandra Mali, Matjaž Perc, Dibakar Ghosh.

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
