## [Decision Letter · Decision Letter 0]

4 Jul 2022

PONE-D-22-17613Eco-evolutionary dynamics of multigames with mutationsPLOS ONE

Dear Dr. Ghosh,

Thank you for submitting your manuscript to PLOS ONE. After careful consideration, we feel that it has merit but does not fully meet PLOS ONE’s publication criteria as it currently stands. Therefore, we invite you to submit a revised version of the manuscript that addresses the points raised during the review process.

ACADEMIC EDITOR: Two reviewers have provided constructive comments. They both agree that the paper is well written and highly appropriate for PloS One audience, and recommend acceptance with very minor amendments. Please carefully take them into account when preparing the revised version.

We look forward to receiving your revised manuscript.

Kind regards,

The Anh Han, Ph.D.

Academic Editor

PLOS ONE

Journal Requirements:

Additional Editor Comments:

Two reviewers have provided constructive comments. They both agree that the paper is well written and highly appropriate for PloS One audience, and recommend acceptance with very minor amendments. Please carefully take them into account when preparing the revised version.

Reviewers' comments:

Reviewer's Responses to Questions

**Comments to the Author**

1. Is the manuscript technically sound, and do the data support the conclusions?

Reviewer #1: Yes

Reviewer #2: Yes

2. Has the statistical analysis been performed appropriately and rigorously? 

Reviewer #1: Yes

Reviewer #2: Yes

3. Have the authors made all data underlying the findings in their manuscript fully available?

Reviewer #1: Yes

Reviewer #2: Yes

4. Is the manuscript presented in an intelligible fashion and written in standard English?

Reviewer #1: Yes

Reviewer #2: Yes

5. Review Comments to the Author

Reviewer #1: In their paper titled "Eco-evolutionary dynamics of multigames with mutations" the authors study an interesting problem, namely what are the options for multigames to yield favorable conditions for cooperation subject to mutations and selfless one-sided contribution of altruist free space. Research reveals a broader range of equilibrium outcomes, which often favor cooperation over defection. With the help of analytical and numerical calculations, the theoretical model the authors are proposing and studying sheds light on the mechanisms that maintain biodiversity, and also explain the evolution of social order in human societies.

Understanding cooperation in social dilemmas is a long standing problem of universal interest across social and natural sciences, and new results in this research field are always well received. In this way, the current manuscript is well-suited for PLOS ONE, and I am sure it will attract citations and readers when published.

I find the manuscript is clear and written with love to detail and care for presentation. The figures are also high quality and should be accessible to the wider audience, as are the results and the research in general. For these reasons I recommend publication in PLOS ONE subject only to minor revision.

The following comments can be considered:

1. Would the approach also work on other social dilemmas games, such that for example the snowdrift and the harmony game would be considered as multigames? And perhaps even more interestingly, would it work in group interactions. The authors can discuss this in the final section if they find of interest.

2. Can the caption to Table 1 be made more comprehensive to say what is the meaning of physical significance and domains.

3. More relevant works, such as [Chaos, Solitons and Fractals 157 (2022) 111987] and [Physica A 591 (2022) 126804], can be cited to make the introduction more comprehensive.

4. The writing is occasional burdened with somewhat awkward grammar and could be polished further before resubmission.

5. Finally, perhaps the authors can make source available to allow others to reproduce the results.

Reviewer #2: The manuscript is generally well written and easy to understand. The authors have proposed a math model that incorporates both prisoner’s dilemma and the snowdrift game. In my opinion, this is quite interesting to see the combination of these two theories. THe authors further extend this model by considering ecological signatures like mutation and selfless one-sided contribution of altruist free space. Their conclusion is that all of these offer a broader range of equilibrium outcomes, and it also often favors cooperation over defection. Their conclusion is well supported by their analytical derviations and numerical calculations. I have some minor queries:

1) How is the steady state in fig 1 being determined?

2) the manuscript is well written, and it will benefit the readers more if there is a section on the limitations and how these limitations can be overcome.

3) I am happy to see that the code has been archived and made available. Can the readers include a section on future work to make the study more comprehensive?

The manuscript can be readily accepted after the above minor queries are addressed.

6. PLOS authors have the option to publish the peer review history of their article (what does this mean?). If published, this will include your full peer review and any attached files.

Reviewer #1: No

Reviewer #2: No

---

## [Author Response · Author response to Decision Letter 0]

22 Jul 2022

Reply to First Reviewer’s commentsPONE-D-22-17613

.

Reviewer #1 (Remarks to the Author):

 In their paper titled "Eco-evolutionary dynamics of multigames with mutations" the authors study an interesting problem, namely what are the options for multigames to yield favorable conditions for cooperation subject to mutations and selfless one-sided contribution of altruist free space. Research reveals a broader range of equilibrium outcomes, which often favor cooperation over defection. With the help of analytical and numerical calculations, the theoretical model the authors are proposing and studying sheds light on the mechanisms that maintain biodiversity, and also explain the evolution of social order in human societies.

Understanding cooperation in social dilemmas is a long standing problem of universal interest across social and natural sciences, and new results in this research field are always well received. In this way, the current manuscript is well-suited for PLOS ONE, and I am sure it will attract citations and readers when published.

I find the manuscript is clear and written with love to detail and care for presentation. The figures are also high quality and should be accessible to the wider audience, as are the results and the research in general. For these reasons I recommend publication in PLOS ONE subject only to minor revision.

Response: We greatly acknowledge the referees for appreciating our work and gratefully acknowledge their insightful suggestions that helped considerably improve the manuscript. As far as the clarifications part is concerned, we have gone through every effort to improve the same in the revised version of the manuscript.

Q.1. Would the approach also work on other social dilemmas games, such that for example the snowdrift and the harmony game would be considered as multigames? And perhaps even more interestingly, would it work in group interactions. The authors can discuss this in the final section if they find of interest.

Response: We acknowledge the referee not only for appreciating our work but also for careful reading and thoughtful comments, which help to improve the manuscript in a significant way. 

The model formed in the manuscript with the help of two different two-person games can be formulated using any two-person games, not necessarily with the prisoner's dilemma and the snowdrift game only. The solution's existence, uniqueness, and boundedness can be similarly guaranteed using the calculations shown in the manuscript for any two different two-person games. We have checked it with two other two-person games, and if the reviewer advises, we will include the generic calculations in the manuscript. However, one should note that the stationary points alter in those cases as different games create a mathematical model with other functions. We strongly believe the role of mutation and the generous free space remain almost similar even for multigame with other two-person games. 

Incorporating group interaction leads to a potentially exciting problem that is out of scope for the present manuscript. We discuss how one can include group interaction by considering interdependent networks and games like the public goods game in the final section of the revised manuscript, as suggested by the anonymous reviewer.

Q.2. Can the caption to Table 1 be made more comprehensive to say what is the meaning of physical significance and domains.

Response: We have enhanced the caption of table 1 by briefly describing the meaning of physical significances and domains.

Q.3 More relevant works, such as [Chaos, Solitons and Fractals 157 (2022) 111987] and [PhysicaA 591 (2022) 126804], can be cited to make the introduction more comprehensive.

Response: The references suggested by the referee concerning the evolution of cooperation in the presence of eco-evolutionary dynamics are indeed noteworthy in the context of our study. We have cited these articles in the discussion section, where a series of relevant publications are introduced.

Q.4. The writing is occasional burdened with somewhat awkward grammar and could be polished further before resubmission.

Response: In the revised manuscript, we have tried our best to improve the grammatical errors throughout our work.

Q.5. Finally, perhaps the authors can make source available to allow others to reproduce the results.

Response: The data that support the findings of this study are openly available online on GitHubathttps://github.com/SayantanNagChowdhury/The-evolutionary-dynamics-in-ecological-multi-games-involving-mutations.

Reply to Second Reviewer’s comments -PONE-D-22-17613

Reviewer #2 (Remarks to the Author):

The manuscript is generally well written and easy to understand. The authors have proposed a math model that incorporates both prisoner’s dilemma and the snowdrift game. In my opinion, this is quite interesting to see the combination of these two theories. THe authors further extend this model by considering ecological signatures like mutation and selfless one-sided contribution of altruist free space. Their conclusion is that all of these offer a broader range of equilibrium outcomes, and it also often favors cooperation over defection. Their conclusion is well supported by their analytical derviations and numerical calculations. I have some minor queries:

Response: We are thankful to the reviewer for appreciating our work and also for valuable suggestions.

Q.1 How is the steady state in fig 1 being determined?

Response: To determine the steady states of Figure 1, we have iterated the constructed model numerically with 2×〖10〗^6 iterations and a fixed integrating step length of 0.01. We used the 4th order Runge–Kutta method (RK4) for the numerical simulation. If the values of x and y remain greater than 0.0001, then we store those values as non-zero values. Otherwise, whenever x and y remain below 0.0001, we consider those values zero. Although we prove the values of x and y remain analytically within the closed interval [0,1], we still maintain that constraint in the numerical code too and store the values only when they stay bounded within this interval [0,1].

Q.2. the manuscript is well written, and it will benefit the readers more if there is a section on the limitations and how these limitations can be overcome.

Response: We greatly acknowledge the referee for appreciating our work. As per your valuable suggestion, we have added a section discussing the limitations and future perspectives. We also try to emphasize how one can overcome those limitations.

Q.3. I am happy to see that the code has been archived and made available. Can the readers include a section on future work to make the study more comprehensive?

Response: We acknowledge the referee not only for appreciating our work but also for careful reading and thoughtful comments, which help to improve the manuscript in a significant way.

We have added a small section in the revised manuscript emphasizing the future works worth studying and need to be addressed in the future.

---

## [Editor Report · Decision Letter 1]

26 Jul 2022

Eco-evolutionary dynamics of multigames with mutations

PONE-D-22-17613R1

Dear Dr. Ghosh,

We’re pleased to inform you that your manuscript has been judged scientifically suitable for publication and will be formally accepted for publication once it meets all outstanding technical requirements.

Kind regards,

The Anh Han, Ph.D.

Academic Editor

PLOS ONE

Additional Editor Comments (optional):

The authors have addressed all the comments from the reviewers very well, and I am happy to recommend its acceptance in the present form.

---

## [Editor Report · Acceptance letter]

30 Jul 2022

PONE-D-22-17613R1 

Eco-evolutionary dynamics of multigames with mutations 

Dear Dr. Ghosh:

I'm pleased to inform you that your manuscript has been deemed suitable for publication in PLOS ONE. Congratulations! Your manuscript is now with our production department. 

Kind regards, 

on behalf of

Dr. The Anh Han 

Academic Editor

PLOS ONE